# Breaking Long-Tailed Learning Bottlenecks: A Controllable Paradigm with Hypernetwork-Generated Diverse Experts

**Zhe Zhao**[1,3], **Haibin Wen**[5], **Zikang Wang**[1], **Pengkun Wang**[1,2]*, **Fanfu Wang**[6],
**Song Lai**[3], **Qingfu Zhang**[3], **Yang Wang**[1,2]*

[1]University of Science and Technology of China (USTC), Hefei, China
[2]Suzhou Institute for Advanced Research, USTC, Suzhou, China
[3]City University of Hong Kong, Hong Kong, China
[4]Harbin Institute of Technology, Harbin, China
[5]MorongAI, Suzhou, China
[6]Lanzhou University, Lanzhou, China
{zz4543@mail.ustc.edu.cn, haibin65535@gmail.com, 2021211940@stu.hit.edu.cn,
wangff21@lzu.edu.cn, songlai2-c@my.cityu.edu.hk, qingfu.zhang@cityu.edu.hk,
pengkun.wang@ustc.edu.cn, angyan@ustc.edu.cn}

## Abstract

Traditional long-tailed learning methods often perform poorly when dealing with inconsistencies between training and test data distributions, and they cannot flexibly adapt to different user preferences for trade-offs between head and tail classes. To address this issue, we propose a novel long-tailed learning paradigm that aims to tackle distribution shift in real-world scenarios and accommodate different user preferences for the trade-off between head and tail classes. We generate a set of diverse expert models via hypernetworks to cover all possible distribution scenarios, and optimize the model ensemble to adapt to any test distribution. Crucially, in any distribution scenario, we can flexibly output a dedicated model solution that matches the user's preference. Extensive experiments demonstrate that our method not only achieves higher performance ceilings but also effectively overcomes distribution shift while allowing controllable adjustments according to user preferences. We provide new insights and a paradigm for the long-tailed learning problem, greatly expanding its applicability in practical scenarios. The code can be found here: https://github.com/DataLab-atom/PRL.

## 1 Introduction

In many real-world tasks such as object detection and image classification, we face the challenge of long-tailed distributions. Since the samples of the head classes account for the vast majority in the datasets while the tail class samples are extremely scarce [21, 16, 6, 17], this extreme imbalance in the data makes the model prone to overfitting towards the head classes during training, resulting in poor performance on the tail classes [34, 8, 30, 20, 39].

To address the long-tailed distribution problem, existing research has proposed a series of methods such as re-sampling [25, 5, 24, 10] and modifying the loss function [17, 6], with the common idea of focusing on improving the performance of the tail classes. However, these methods typically assume that the distributions of the training and test data remain invariant, and thus cannot well handle the

---

*Pengkun Wang and Yang Wang are corresponding authors.

38th Conference on Neural Information Processing Systems (NeurIPS 2024).

common situations of distribution shift between training and testing in real-world scenarios. Some more recent works such as RIDE [32] and LADE [12] propose using multiple expert models to obtain stronger distribution adaptability. Building on this, SADE [38] further adaptively combines the outputs of these experts during testing to adapt to the current test distribution. These approaches alleviate the problem of distribution mismatch between training and testing to some extent [23].

However, in addressing distribution shifts across different test scenarios, the goal of these multi-expert model-based methods is still to maximize the overall performance, i.e., pursuing the optimal overall performance metric across all classes, and obtaining a fixed trade-off for this purpose[13, 40]. But in different application scenarios, *users may have different preferences and needs for the relative trade-off between head and tail classes. Simply pursuing the overall optimal solution may not meet this flexibility requirement*[17, 35, 43]. For example, in classifying lung CT images, when screening for difficult cases, we care more about whether all possible disease types (i.e., tail classes) can be covered to avoid missed diagnoses, compared to routine physical examinations. For some serious diseases such as lung cancer, we may also be willing to moderately increase the false positive rate in exchange for higher coverage of the tail classes, to ensure that no patients are missed. Another example is wildlife detection. Within nature reserves, we want the model to accurately detect common species (i.e., head classes) to understand their population sizes. But when looking for rare species (i.e., tail classes), we care more about covering all species, even at the cost of some false detections.

As can be seen, in different application scenarios, there are significant differences in user preferences for the weighting of head and tail categories, which current long-tailed learning methods often fail to fully satisfy. Therefore, developing an interpretable and controllable method for handling long-tail distributions that adapts to specific user preferences for head and tail categories becomes a new research direction in the field of long-tailed learning.

In light of this, we propose an inter**p**retable and cont**r**ollable **l**ong-tail learning method (**PRL**). This method aims not only to overcome potential distribution shifts from a single training distribution to any testing distribution but more importantly, to flexibly adjust the weights of head and tail categories according to actual user demands. To address these challenges, we introduce a new long-tailed learning paradigm based on a diverse set of experts and hypernetworks, which can adapt to a wide range of distribution scenarios and meet personalized user preferences.

To tackle the aforementioned challenges, we propose a new long-tailed learning paradigm based on diverse experts and hypernetworks, as illustrated in Figure. For the first challenge, existing multi-expert model-based methods train fixed expert models for specific distributions, requiring strong distribution assumptions and struggling to handle more complex and variable distributions. Therefore, instead of maximizing the performance of each expert individually, we pursue modeling and optimizing the hypervolume over the entire Pareto front curve, learning a set of solutions that cover all possible distribution scenarios. This requires us to sample with the goal of covering the entire Pareto front during optimization. For the second challenge, unlike LADE and SADE which output a fixed trade-off solution under distribution shift, we can flexibly output a dedicated model solution that matches the user's preference in any test distribution scenario. In this way, our method can not only adapt to changes in the test distribution, but also allow controllable adjustment of the head-tail trade-off according to the user's actual needs.

Our contributions can be summarized as follows:

- *New scenario and insight*: we make the first attempt at a controllable trade-off based on user preferences in the context of long-tailed learning and test distribution shift scenarios, greatly expanding the applicability in real-world scenarios.

- *New learning paradigm*: we propose a new interpretable and controllable long-tailed learning method that can acquire the ability to overcome test distribution shift from a single distribution dataset and satisfy user preferences in any shifted distribution scenario.

- *Compelling empirical results*: extensive experiments demonstrate that our method achieves higher performance ceilings, effectively overcomes test distribution shift, and can be controlled by user preferences.

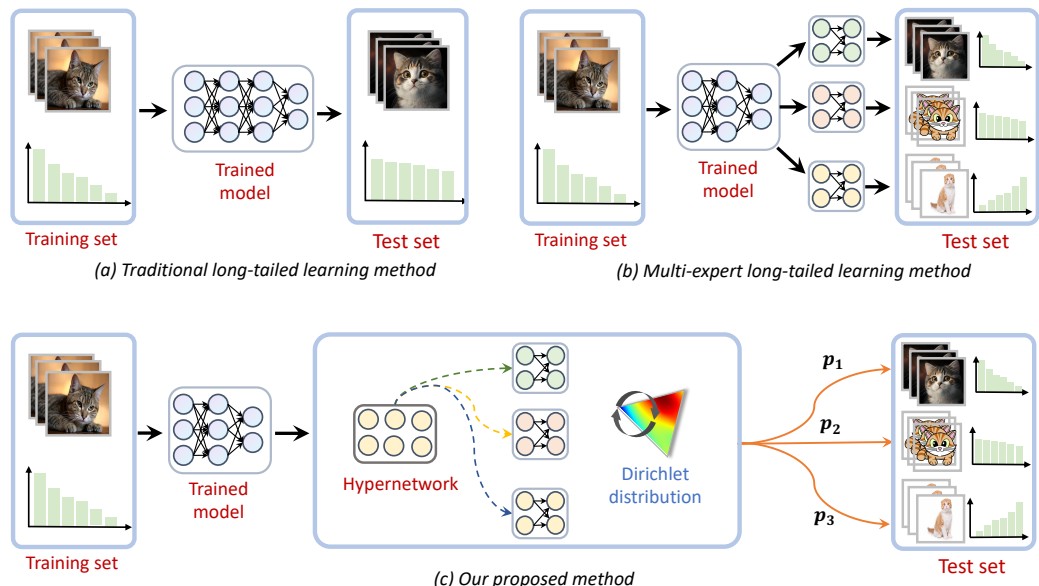

Figure 1: Illustration of our method: (a) Existing methods train for a specific long-tailed distribution but may fail on arbitrarily skewed test distributions. (b) Multi-expert learns different experts for different distributions from one training set but lacks flexibility for arbitrary distributions/preferences. (c) Our method samples preference vectors during training to simulate distributions, and can flexibly adjust the preference vector during testing for flexible long-tailed classification.

## 2 Related Work

Long-tailed distributions are prevalent in real-world data, leading to imbalanced datasets that pose challenges for machine learning models [30, 20]. To address this issue, researchers have proposed various methods, including re-sampling, loss function modification, and multi-expert models.

Re-sampling methods balance class distributions by oversampling tail classes [24, 3] or undersampling head classes [7]. Loss function modification approaches assign higher weights to tail class losses [27, 26] or use meta-learning to alleviate undersampling issues [14, 32]. Multi-expert models train multiple experts on different class distributions and combine their outputs, adapting to various test distributions [37, 38, 31]. Most existing methods assume specific distributions during training or testing, limiting real-world applicability with distribution shifts, and cannot accommodate varying user needs for head and tail class trade-offs. We propose an approach to overcoming distribution assumptions and achieve interpretable, controllable trade-offs in long-tailed learning.

## 3 Theory

In this section, we analyze the distribution shift problem from a theoretical perspective and provide the definition and properties of the environment's total variation distance, laying the theoretical foundation for the methods section.

**Definition 1** (Distribution Discrepancy across Environments). *Given $M$ training environments $\mathcal{E}_1, \ldots, \mathcal{E}_M$, with class prior probability vectors $\boldsymbol{\pi}^1, \ldots, \boldsymbol{\pi}^M$, respectively, where $\boldsymbol{\pi}^m = (\pi_1^m, \ldots, \pi_K^m)$, $\pi_K^m$ denotes the probability of the $k$-th class appearing in environment $\mathcal{E}_m$, and $\sum_{k=1}^{K} \pi_k^m = 1$. If there exist $i, j, k, l$ such that $\frac{\pi_k^i}{\pi_l^i} \neq \frac{\pi_k^j}{\pi_l^j}$ holds, then these $M$ environments are said to have distribution discrepancy.*

Traditional empirical risk minimization (ERM) methods on a single training distribution struggle to handle distribution discrepancy, which can affect generalization. This limitation can be characterized by the following theorem:

**Theorem 1** (Limitation of ERM). *Let $f(\boldsymbol{x}; \boldsymbol{\theta})$ be a classifier learned via ERM on $\mathcal{E}_m$, then its risk on the test environment $\mathcal{E}_{test}$ is:*

$$R_{test}(f) = R_m(f) + \sum_{i=1}^{K} (\pi_i^{test} - \pi_i^m) \cdot \mathbb{E}_{\boldsymbol{x} \sim P_{test}(\boldsymbol{x}|y=i)}[\ell(f(\boldsymbol{x}; \boldsymbol{\theta}), i)] \tag{1}$$

*where $R_m(f)$ and $R_{test}(f)$ are the risks of $f$ on $\mathcal{E}_m$ and $\mathcal{E}_{test}$, respectively, and $\boldsymbol{\pi}^{test}$ is the class prior of the test environment.*

To measure the distribution discrepancy across environments, we introduce the environment's total variation distance (ETVD):

**Definition 2** (Environment Total Variation Distance). *The total variation distance between environments $\mathcal{E}_i$ and $\mathcal{E}_j$ is defined as: $\delta(\mathcal{E}_i, \mathcal{E}_j) = \frac{1}{2} \sum_{k=1}^{K} |\pi_k^i - \pi_k^j|$, and the ETVD of $M$ environments is defined as: $\Delta(\mathcal{E}_1, \ldots, \mathcal{E}_M) = \max_{i,j \in \{1, \ldots, M\}} \delta(\mathcal{E}_i, \mathcal{E}_j)$*

Using ETVD, we can further bound the risk of the ERM-learned classifier on the test environment:

**Corollary 1.** *Under the assumptions of Theorem 1, let $M = \max_{i,\boldsymbol{x}} \ell(f(\boldsymbol{x}; \boldsymbol{\theta}), i)$, then*

$$R_{test}(f) \leq R_m(f) + 2M \cdot (\delta(\mathcal{E}_m, \mathcal{E}_{test}) + \Delta(\mathcal{E}_1, \ldots, \mathcal{E}_M)) \tag{2}$$

This corollary shows that the test risk of the ERM-learned classifier is affected not only by the distribution discrepancy between the training environment and the test environment but also by the distribution discrepancy among training environments (i.e., ETVD). To overcome the diversity shift, we propose minimizing the empirical risks across multiple training environments to capture the distributional characteristics of different environments, thereby learning a set of diversity experts.

Next, we provide a theoretical analysis of the domain adaptation algorithm based on diversity experts proposed in this paper. To characterize the generalization performance of this algorithm, we first introduce the following notations:

Let $\{f_1, \ldots, f_N\}$ be the $N$ experts learned via ERM on the $N$ training environments $\{\mathcal{E}_1, \ldots, \mathcal{E}_N\}$, respectively, and $\hat{f}$ be the final classifier obtained by ensembling these $N$ experts. Define the empirical risk of the ensemble classifier $\hat{f}$ on environment $\mathcal{E}_m$ as:

$$\hat{R}_m(\hat{f}) = \frac{1}{N} \sum_{i=1}^{N} R_m(f_i) \tag{3}$$

We can obtain the following theorem regarding the generalization performance of the ensemble classifier:

**Theorem 2.** *Under the above notations and definitions, the risk of the ensemble classifier $\hat{f}$ on the test environment $\mathcal{E}_{test}$ satisfies:*

$$R_{test}(\hat{f}) \leq \frac{1}{N} \sum_{m=1}^{N} R_m(f_m) + 2M \cdot \left( \frac{1}{N} \sum_{m=1}^{N} \delta(\mathcal{E}_m, \mathcal{E}_{test}) + \frac{N-1}{N} \Delta(\mathcal{E}_1, \ldots, \mathcal{E}_N) \right) \tag{4}$$

*where $M = \max_{i,\boldsymbol{x}} \ell(\hat{f}(\boldsymbol{x}), i)$.*

Theorem 2 shows that the test risk of the ensemble classifier consists of three parts: the average empirical risk of all experts, the average total variation distance between the training environments and the test environment, and the weighted average of ETVD among the training environments. Compared to single-environment ERM, the diversity experts method learns a set of experts to capture the distributional characteristics of different environments, which can reduce the distribution discrepancy between the training environments and the test environment, thereby achieving better generalization performance.

## 4 Methodology

### 4.1 Problem Formulation

Consider a $K$-class classification problem with a training set $\mathcal{D} = \{(\boldsymbol{x}_i, y_i)\}_{i=1}^{N}$, where each class $k$ has $N_k$ samples. Let $\mathcal{P}_{\text{train}}$ denote the empirical distribution over $\mathcal{D}$. The goal is to learn a classifier

$f : \mathcal{X} \to \mathbb{R}^K$ that generalizes well across various test distributions $\mathcal{P}_{\text{test}}$. Traditional empirical risk minimization (ERM) methods optimize the loss under $\mathcal{P}_{\text{train}}$, but may fail to adapt to changes in $\mathcal{P}_{\text{test}}$, especially in long-tailed scenarios.

To improve the robustness of $f$, we optimize the losses under multiple importance-weighted distributions. Define an $M$-dimensional simplex:

$$\Delta_M := \left\{ \boldsymbol{\alpha} \in \mathbb{R}_+^M \mid \sum_{i=1}^{M} \alpha_i = 1 \right\} \tag{5}$$

Each $\boldsymbol{\alpha} \in \Delta_M$ corresponds to an importance-weighted distribution $\mathcal{P}_{\boldsymbol{\alpha}}$:

$$\mathcal{P}_{\boldsymbol{\alpha}}(\boldsymbol{x}, y) := \sum_{k=1}^{K} \alpha_k \cdot \mathcal{P}_k(\boldsymbol{x} \mid y) \cdot \mathcal{P}_k(y) \tag{6}$$

where $\mathcal{P}_k(\boldsymbol{x} \mid y)$ and $\mathcal{P}_k(y) = \frac{N_k}{N}$ are the conditional distribution and prior for class $k$, respectively.

The objective is to learn a set of classifiers $\mathcal{F} := \{f^{(i)}\}_{i=1}^{M}$ that achieve low risk simultaneously across all $\mathcal{P}_{\boldsymbol{\alpha}}$, forming the Pareto optimal solution:

$$\min_{\mathcal{F}} \left( \mathcal{R}_{\mathcal{P}_{\boldsymbol{\alpha}_1}}(\mathcal{F}), \ldots, \mathcal{R}_{\mathcal{P}_{\boldsymbol{\alpha}_M}}(\mathcal{F}) \right) \tag{7}$$

where $\mathcal{R}_{\mathcal{P}_{\boldsymbol{\alpha}}}(\mathcal{F}) := \mathbb{E}_{(\boldsymbol{x}, y) \sim \mathcal{P}_{\boldsymbol{\alpha}}} \left[ \frac{1}{M} \sum_{i=1}^{M} \ell(f^{(i)}(\boldsymbol{x}), y) \right]$. Pursuing an approximate Pareto solution across all distributions leads to models with stronger generalization capabilities.

## 4.2 Diverse Experts

Let $\mathcal{X}$ and $\mathcal{Y}$ denote the input and output spaces, respectively. We introduce $T = 3$ classifiers $\{f_i\}_{i=1}^{T}$ as diverse experts. These experts share a feature extractor $\phi_\theta : \mathcal{X} \to \mathbb{R}^d$, but use different classifier heads $\{g_{w_i}\}_{i=1}^{T}$:

$$f_i(\mathbf{x}) = g_{w_i}(\phi_\theta(\mathbf{x})), \quad i = 1, \ldots, T \tag{8}$$

To generate diverse experts, we introduce a hypernetwork $h_\psi$ that takes random noise $\mathbf{z} \in \mathbb{R}^k$ as input and outputs the classifier head parameters $\mathbf{w}_i \in \mathbb{R}^{d \times C}$:

$$\mathbf{w}_i = h_\psi(\mathbf{z}_i), \quad \mathbf{z}_i \sim \text{Dir}(\boldsymbol{\alpha}), \quad i = 1, \ldots, T \tag{9}$$

where $\text{Dir}(\boldsymbol{\alpha})$ is the Dirichlet distribution with parameter $\boldsymbol{\alpha} \in \mathbb{R}_+^k$. The hypernetwork $h_\psi$ consists of three linear layers with ReLU activations.

During training, we sample $\{\mathbf{z}_i\}_{i=1}^{T}$ from $\text{Dir}(\boldsymbol{\alpha})$ and use $h_\psi$ to generate $\{\mathbf{w}_i\}_{i=1}^{T}$. The loss function for a training batch is:

$$\mathcal{L} = \sum_{i=1}^{T} \mathcal{L}_i(f_i) \tag{10}$$

where $\mathcal{L}_i$ is the classification loss for the $i$-th expert $f_i$, defined the same as in SADE: $\mathcal{L}_1$ is the standard cross-entropy loss; $\mathcal{L}_2$ is the balanced softmax loss, where the logits are adjusted by adding the log of the prior probabilities of each class; $\mathcal{L}_3$ is the inverse softmax loss, where the logits are adjusted by adding the log of the prior probabilities and subtracting the scaled log of the inverse prior probabilities.

## 4.3 Stochastic Convex Ensemble

Let $\mathcal{L}_i(\boldsymbol{\Theta}, \mathcal{D})$ denote the loss function of the $i$-th expert $f_i$ on dataset $\mathcal{D}$, where $\boldsymbol{\Theta} = \{\theta, \psi\}$ represents all trainable parameters. The objective is to jointly optimize the losses of all $T$ experts:

$$\min_{\boldsymbol{\Theta}} \sum_{i=1}^{T} \mathcal{L}_i(\boldsymbol{\Theta}, \mathcal{D}) \tag{11}$$

To promote diversity among experts, we introduce the Stochastic Convex Ensemble (SCE) strategy, which aims to minimize the worst-case loss of the convex combination of experts:

$$\min_{\boldsymbol{\Theta}} \max_{\mathbf{p} \in \Delta_T} \sum_{i=1}^{T} p_i \mathcal{L}_i(\boldsymbol{\Theta}, \mathcal{D}) \tag{12}$$

where $\mathbf{p} = (p_1, \cdots, p_T)^\top \in \Delta_T$ is the weight vector, and $\Delta_T := \{\mathbf{p} \in \mathbb{R}_+^T | \sum_{i=1}^{T} p_i = 1\}$ is the $T$-dimensional simplex.

Inspired by the max-min inequality, we relax the SCE objective to:

$$\min_{\boldsymbol{\Theta}} \left( \sum_{i=1}^{T} \mathcal{L}_i(\boldsymbol{\Theta}, \mathcal{D}) + \lambda \cdot \log \sum_{i=1}^{T} \exp\left( \frac{1}{\lambda} \mathcal{L}_i(\boldsymbol{\Theta}, \mathcal{D}) \right) \right) \tag{13}$$

where $\lambda > 0$ is a hyperparameter. As $\lambda \to 0$, the relaxed objective approaches the original SCE objective. The term $\lambda \cdot \log \sum_{i=1}^{T} \exp\left( \frac{1}{\lambda} \mathcal{L}_i(\boldsymbol{\Theta}, \mathcal{D}) \right)$ promotes diversity among experts.

### 4.4 Preference-Controlled Trade-off

During testing, we can control the trade-off between head and tail classes using a preference vector $\boldsymbol{\alpha}^* = (\alpha_1^*, \alpha_2^*, \alpha_3^*)^\top \in \Delta^3$, where $\Delta^3$ is the 3-dimensional simplex.

Given a trained preference vector $\boldsymbol{r} = (r_1, r_2, r_3)^\top \in \Delta^3$, we compute the test-time preference vector $\hat{\boldsymbol{r}} \in \Delta^3$ as:

$$\hat{\boldsymbol{r}} = \frac{\boldsymbol{r} \odot \boldsymbol{\alpha}^*}{\boldsymbol{r}^\top \boldsymbol{\alpha}^*} \tag{14}$$

where $\odot$ denotes the Hadamard product. The test-time preference vector $\hat{\boldsymbol{r}}$ is then input to the hypernetwork $h_\psi$ to generate the classifier head parameters for each expert:

$$\hat{\boldsymbol{W}}_i = h_\psi(\hat{\boldsymbol{r}}), \quad i = 1, \cdots, T \tag{15}$$

where $\hat{\boldsymbol{W}}_i \in \mathbb{R}^{d \times C}$ is the weight matrix for the $i$-th expert's classifier head. For a test sample with feature vector $\boldsymbol{x} \in \mathbb{R}^d$, the output of the $i$-th expert is:

$$\hat{\boldsymbol{y}}_i = \begin{cases} \dfrac{\boldsymbol{x}^\top}{\|\boldsymbol{x}\|_2} \cdot \dfrac{\hat{\boldsymbol{W}}_i}{\|\hat{\boldsymbol{W}}_i\|_\mathrm{F}}, & \text{if normalized} \\ \boldsymbol{x}^\top \hat{\boldsymbol{W}}_i + \hat{\boldsymbol{b}}_i^\top, & \text{otherwise} \end{cases} \tag{16}$$

where $\|\cdot\|_\mathrm{F}$ is the Frobenius norm and $\hat{\boldsymbol{b}}_i \in \mathbb{R}^C$ is the bias vector for the $i$-th expert. By adjusting $\boldsymbol{\alpha}^*$, we can control the model's focus on head or tail classes, enabling flexible trade-offs to suit different application needs.

**An observation on our method.** To better understand this part, we use Figure 2 to demonstrate the effectiveness of preference control in overcoming distribution shifts, as well as the flexibility of our method. For preferences, the coordinate system is a three-dimensional orthogonal coordinate; for accuracy, the coordinate system represents the performance on the farward50, uni., and backward50 splits of the CIFAR100-LT dataset. The dark plane represents the plane formed by different preference vectors, and the outer surface represents the corresponding performance on the three distributions for these preference vectors. The yellow dots are the results of running SADE, whose preferences are uncontrollable, so the results of each run are random dots, lying below our purple plane, indicating that their performance is lower than our method (i.e., being dominated in the Pareto optimal set). This figure illustrates that our method can cover unknown distributions without additional training, and unlike previous methods, it can trade off performance by adjusting the preference vector. We will analyze this in more depth in the experimental section.

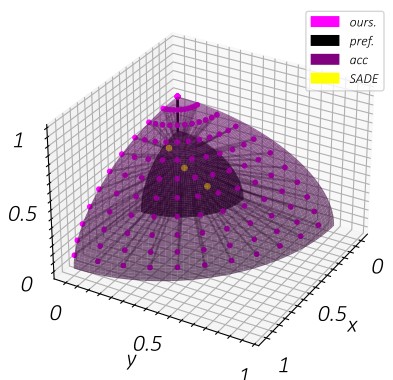

Figure 2: Mapping from preference to model properties.

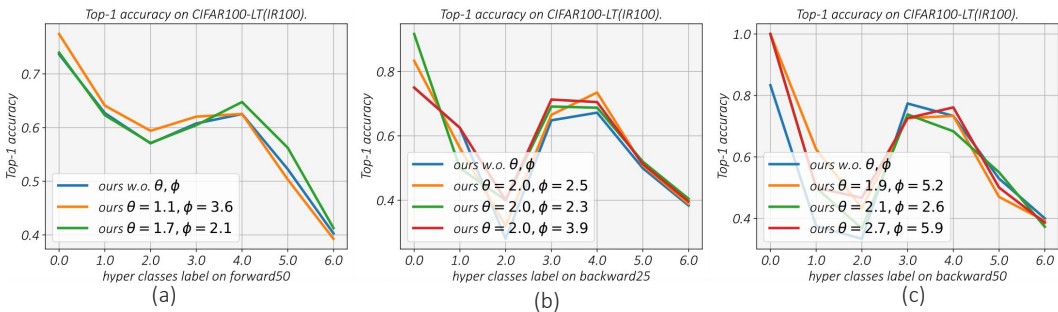

Figure 3: Analysis of the preference control for the trade-off between head and tail class performance. We present the results on three distributions. The vertical axis represents accuracy. The horizontal axis shows the results after clustering by frequency, from head classes to tail classes (left to right).

## 5   Experiments

In this section, we first evaluate the superiority of PRL in terms of both standard and test-agnostic long-tailed recognition to demonstrate that our method has a higher performance ceiling under the traditional setup. Then, we analyze the effectiveness of our method in changing the trade-off for long-tailed classes through input preferences. Furthermore, we conduct necessary ablation studies.

### 5.1   Experimental Setups

**Datasets.** We evaluate our method on four benchmark datasets: ImageNet-LT [20], CIFAR100-LT [4], Places-LT [20], and iNaturalist 2018 [29]. These datasets have varying imbalance ratios, ranging from 10 to 256. CIFAR100-LT has three versions with different imbalance ratios. Detailed statistics are in Appendix D.

**Baselines.** We compare PRL with various state-of-the-art long-tailed recognition methods, including two-stage methods (MiSLAS [41]), logit-adjusted training (Balanced Softmax [15], LADE [12]), ensemble learning (RIDE [32], SADE [38]), causal inference (Causal [28]), representation learning (LSC [33]), and balanced posterior averaging (BalPoE [1]). These methods address the long-tail problem from different perspectives. Further details are provided in the appendixA.

**Evaluation protocols and implementation details.** We evaluate the models on multiple test datasets with different class distributions using micro accuracy. We report the accuracy of many-shot, medium-shot, and few-shot classes. We use the same setup for all methods, including ResNeXt-50 for ImageNet-LT, ResNet-32 for CIFAR100-LT, ResNet-152 for Places-LT, and ResNet-50 for iNaturalist 2018 as backbones. We employ hypernets (MLPs) to output trainable parameters of experts and adopt the cosine classifier for prediction. Unless specified, we use $\alpha = 1.2$ for the Dirichlet distribution, $\mu = 0.3$ for stochastic annealing, SGD with momentum 0.9, train for 200 epochs, and set the initial learning rate to 0.1 with linear decay. During test-time training, we train aggregation weights for 5 epochs with a batch size of 128, using the same optimizer and learning rate as in training. Other details please refer to Appendix G.

### 5.2   Comparative Evaluation on Standard and Test-Agnostic Long-Tailed Recognition

We conduct extensive experiments on four widely-used long-tailed datasets, including CIFAR100-LT, Places-LT, iNaturalist 2018, and ImageNet-LT, to evaluate the performance of our proposed PRL method in comparison with state-of-the-art approaches.

**Results on standard long-tailed recognition.** Table 1 demonstrates the effectiveness of our proposed method, PRL, on four benchmark datasets under the standard long-tailed recognition setting, where the test class distribution is uniform. PRL consistently achieves the highest top-1 accuracy across all datasets, outperforming the previous state-of-the-art methods, LSC [33] and BalPoE [1]. On CIFAR100-LT, PRL improves the accuracy by 0.6% to 0.8% compared to LSC and BalPoE, showcasing its robustness to different imbalance ratios (IR=10, 50, and 100). Similarly, on Places-

Table 1: Top-1 accuracy on CIFAR100-LT, Places-LT, iNaturalist 2018, and ImageNet-LT, where the test class distribution is uniform.

| Method | CIFAR100-LT | | | Places-LT | iNaturalist 2018 | ImageNet-LT |
|---|---|---|---|---|---|---|
| | IR=10 | IR=50 | IR=100 | | | |
| Softmax | 59.1 | 45.6 | 41.4 | 31.4 | 64.7 | 48.0 |
| Causal [28] | 59.4 | 48.8 | 45.0 | 32.2 | 64.4 | 50.3 |
| Balanced Softmax [15] | 61.0 | 50.9 | 46.1 | 39.4 | 70.6 | 52.3 |
| MiSLAS [41] | 62.5 | 51.5 | 46.8 | 38.3 | 70.7 | 51.4 |
| LADE [12] | 61.6 | 50.1 | 45.6 | 39.2 | 69.3 | 52.3 |
| RIDE [32] | 61.8 | 51.7 | 48.0 | 40.3 | 71.8 | 56.3 |
| SADE [38] | 63.6 | 53.8 | 48.8 | 40.9 | 72.7 | 58.8 |
| LSC [33] | 65.0 | 56.5 | 51.8 | 41.3 | 73.9 | 60.2 |
| BalPoE [1] | 64.8 | 56.3 | 52.0 | 40.8 | 75.0 | 59.3 |
| PRL(ours) | **65.6** | **57.3** | **52.8** | **41.6** | **75.1** | **60.8** |

Table 2: Top-1 accuracy on CIFAR100-LT (IR100) with various unknown test class distributions.

| Method | Prior | Forward-LT | | | | | Uni. | Backward-LT | | | | |
|---|---|---|---|---|---|---|---|---|---|---|---|---|
| | | 50 | 25 | 10 | 5 | 2 | 1 | 2 | 5 | 10 | 25 | 50 |
| Softmax | ✗ | 63.3 | 62.0 | 56.2 | 52.5 | 46.4 | 41.4 | 36.5 | 30.5 | 25.8 | 21.7 | 17.5 |
| BS | ✗ | 57.8 | 55.5 | 54.2 | 52.0 | 48.7 | 46.1 | 43.6 | 40.8 | 38.4 | 36.3 | 33.7 |
| MiSLAS | ✗ | 58.8 | 57.2 | 55.2 | 53.0 | 49.6 | 46.8 | 43.6 | 40.1 | 37.7 | 33.9 | 32.1 |
| LADE | ✗ | 56.0 | 55.5 | 52.8 | 51.0 | 48.0 | 45.6 | 43.2 | 40.0 | 38.3 | 35.5 | 34.0 |
| LADE | ✓ | 62.6 | 60.2 | 55.6 | 52.7 | 48.2 | 45.6 | 43.8 | 41.1 | 41.5 | 40.7 | 41.6 |
| RIDE | ✗ | 63.0 | 59.9 | 57.0 | 53.6 | 49.4 | 48.0 | 42.5 | 38.1 | 35.4 | 31.6 | 29.2 |
| SADE | ✗ | 65.2 | 62.5 | 58.8 | 55.4 | 51.2 | 48.8 | 43.0 | 43.9 | 42.4 | 42.2 | 42.0 |
| LSC | ✗ | 67.8 | 64.2 | 60.2 | 58.1 | 53.2 | 51.6 | 44.7 | 45.7 | 44.2 | 44.7 | 48.0 |
| BalPoE | ✗ | 69.0 | 65.2 | 61.2 | 59.0 | 54.2 | 51.7 | 45.7 | 46.6 | 45.2 | 45.2 | 45.8 |
| PRL (ours) | ✗ | **69.5** | **65.7** | **61.7** | **59.5** | **54.7** | **52.2** | **46.2** | **47.1** | **45.7** | **45.7** | **48.5** |

LT, iNaturalist 2018, and ImageNet-LT, PRL obtains the best performance, surpassing the existing methods by a clear margin. The superior performance of PRL in the standard long-tailed recognition setting validates the efficacy of our approach in mitigating the bias towards head classes and improving the recognition accuracy of tail classes.

**Results on distribution-shift long-tailed recognition.** We evaluate the performance of PRL and other methods in the distribution-shift long-tailed recognition setting, where the test class distribution is unknown and different from the training distribution. Tables 2 and 3 show the Top-1 accuracy results on various test class distributions (including forward LT, uniform, and backward LT) for CIFAR100-LT (IR=100) and ImageNet-LT, respectively.

On different test distributions of both datasets, PRL consistently outperforms all compared methods. On CIFAR100-LT (IR100), PRL achieves the highest accuracy across all settings, surpassing LSC [33], BalPoE [1], and SADE [38]. Even under the most challenging backward LT distribution, PRL can maintain its outstanding performance. On ImageNet-LT, PRL obtains the best results across all test distributions, significantly outperforming LSC, BalPoE, and SADE.

The consistent improvements achieved by PRL highlight its higher performance ceiling, indicating the effectiveness of our method design in overcoming distribution shifts. We further conduct distribution-shift experiments on Places-LT and iNaturalist 2018, where PRL also achieves impressive results. Please refer to the AppendixF for detailed results.

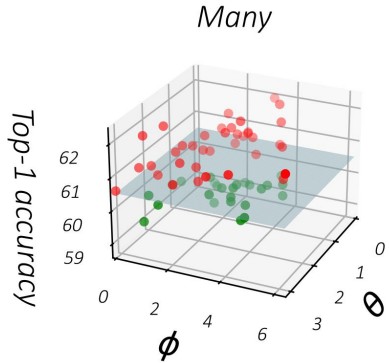

Figure 4: A more comprehensive example of how preference influences performance.

Table 3: Top-1 accuracy on ImageNet-LT with various unknown test class distributions.

| Method | Prior | Forward-LT | | | | | Uni. | Backward-LT | | | | |
|---|---|---|---|---|---|---|---|---|---|---|---|---|
| | | 50 | 25 | 10 | 5 | 2 | 1 | 2 | 5 | 10 | 25 | 50 |
| Softmax | ✗ | 66.1 | 63.8 | 60.3 | 56.6 | 52.0 | 48.0 | 43.9 | 38.6 | 34.9 | 30.9 | 27.6 |
| BS | ✗ | 63.2 | 61.9 | 59.5 | 57.2 | 54.4 | 52.3 | 50.0 | 47.0 | 45.0 | 42.3 | 40.8 |
| MiSLAS | ✗ | 61.6 | 60.4 | 58.0 | 56.3 | 53.7 | 51.4 | 49.2 | 46.1 | 44.0 | 41.5 | 39.5 |
| LADE | ✗ | 63.4 | 62.1 | 59.9 | 57.4 | 54.6 | 52.3 | 49.9 | 46.8 | 44.9 | 42.7 | 40.7 |
| LADE | ✓ | 65.8 | 63.8 | 60.6 | 57.5 | 54.5 | 52.3 | 50.4 | 48.8 | 48.6 | 49.0 | 49.2 |
| RIDE | ✗ | 67.6 | 66.3 | 64.0 | 61.7 | 58.9 | 56.3 | 54.0 | 51.0 | 48.7 | 46.2 | 44.0 |
| SADE | ✗ | 69.7 | 67.5 | 65.4 | 62.3 | 60.3 | 58.3 | 56.7 | 54.9 | 54.3 | 53.1 | 52.6 |
| LSC | ✗ | 72.0 | 69.7 | 67.5 | 65.3 | 62.7 | 60.2 | 59.2 | 58.5 | 57.9 | 57.5 | 57.0 |
| BalPoE | ✗ | 72.2 | 69.7 | 67.2 | 64.3 | 62.2 | 59.5 | 58.5 | 57.7 | 56.9 | 56.7 | 56.6 |
| PRL (ours) | ✗ | **72.7** | **70.2** | **68.0** | **65.8** | **63.2** | **60.7** | **59.7** | **59.0** | **58.4** | **58.0** | **57.5** |

Table 4: Control of trade-off preference for long-tailed classes with different preferences, **bold text**, underlined text, and dashed underline respectively indicate the highest performance of the head, middle, and tail classes in this line.

| Dist. | | R=(1.0, 2.7) | | | R=(0.5, 2.5) | | | R=(1.9, 1.1) | | |
|---|---|---|---|---|---|---|---|---|---|---|
| | | Many | Middle | Few | Many | Middle | Few | Many | Middle | Few |
| Forward | 50 | **61.4** | 50.4 | 36.5 | 61.0 | 52.6 | 31.5 | 61.1 | 48.9 | 40.3 |
| | 25 | **61.6** | 48.3 | 28.4 | 60.6 | 49.6 | 31.5 | 59.7 | 49.4 | 33.1 |
| Uni | 1 | 61.6 | 51.4 | 33.2 | 61.6 | 51.5 | 33.2 | 61.6 | 51.4 | 33.2 |
| Backward | 25 | **63.8** | 49.4 | 31.1 | 60.2 | 48.2 | 32.1 | 63.2 | 48.2 | 32.2 |
| | 50 | **66.6** | 47.1 | 30.6 | 66.1 | 48.9 | 30.9 | 64.6 | 47.8 | 31.7 |

**User preference control.** We evaluated the model's performance on many-shot, medium-shot, and few-shot classes on CIFAR100-LT under different preference settings (R=(1.0, 2.7), R=(0.5, 2.5), R=(1.9, 1.1)). Table 4 shows that by adjusting the preference value R, we can effectively control the trade-off between many-shot and few-shot classes. When R=(1.0, 2.7), the model performs best on many-shot classes; when R=(1.9, 1.1), the model performs better on few-shot classes at the cost of a slight drop in performance on many-shot classes. R=(0.5, 2.5) achieves the best performance on medium-shot classes, indicating that our method can balance performance across different classes with an appropriate setting. As shown in Figure 3, we analyze the performance trade-offs between head and tail classes across three different distributions, demonstrating how our preference control mechanism allows flexible adjustment of model behavior. The results clearly show how adjusting preferences affects accuracy across different class frequency groups.

Figure 4 provides a more comprehensive visualization of the performance on the head classes under the forward50 distribution. The plane represents the performance of the head classes without inputting any preference, while the red dots indicate the preference positions in polar coordinates that can improve the performance of the head classes, and the green dots represent the preference positions that may degrade the performance. These experimental results demonstrate the effectiveness of our method in controlling the trade-off for long-tailed classes based on user preferences. By adjusting the preference without the need for retraining the model, we can flexibly adapt to different application

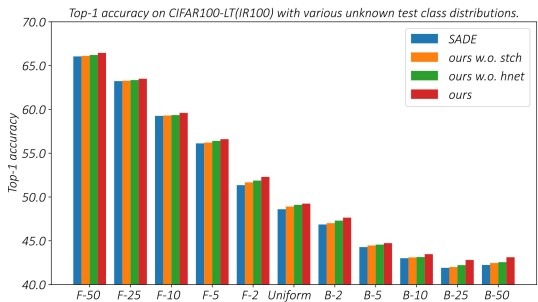

Figure 5: Ablation analysis, including the ablation of the hypernetwork and Chebyshev polynomials.

scenarios and requirements, achieving a desired trade-off in long-tailed recognition tasks that aligns with practical needs.

**Ablation study.** We conduct ablation studies on CIFAR100-LT to evaluate the impact of removing the hypernetwork (w.o. hnet) and removing the Chebyshev polynomial (w.o. stch) on the model's performance under different unknown test class distributions (as shown in Figure 5). The complete model (ours) performs best across all distributions. Removing either the hypernetwork or the Chebyshev polynomial leads to performance degradation, highlighting their importance in dynamically adjusting the model behavior to adapt to distribution shifts and learning preference-aware representations. This ablation study verifies the effectiveness of different components in our method, which work together to better handle unknown test distributions and data imbalance issues in long-tailed recognition.

# 6 Conclusion

This study introduces a novel long-tailed learning paradigm to address distribution shifts between training and testing datasets. Our hypernetwork-based approach generates adaptable classifiers, achieving Pareto optimality for real-time adaptation. During inference, the model adjusts based on user-defined trade-offs between head and tail classes, enhancing flexibility. Empirical results show improved accuracy and adaptability to class imbalances and distribution shifts. Our work establishes an interpretable, generalizable, and controllable framework for long-tailed learning, meeting diverse user needs.

# Acknowledgements

This work was supported by the Natural Science Foundation of China Youth Project (No. 62402472), the Natural Science Foundation of Jiangsu Province of China Youth Project (No. BK20240461), the Research Grants Council of the Hong Kong Special Administrative Region, China (GRF Project No. CityU 11215723), National Natural Science Foundation of China (No.62072427, No.12227901), the Project of Stable Support for Youth Team in Basic Research Field, CAS (No.YSBR-005), and Academic Leaders Cultivation Program, USTC.

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

# Appendix
# Breaking Long-Tailed Learning Bottlenecks:
# A Controllable Paradigm with Hypernetwork-Generated Diverse Experts

## A  Baselines Details

In this section, we provide a comprehensive overview of some state-of-the-art methods for long-tailed recognition, which will serve as baselines for comparison with our proposed PRL approach.

- **Two-stage methods** decouple representation learning and classifier training to mitigate the bias towards head classes. MiSLAS [41] introduces a mixup-based strategy in the second stage to enhance the learning of tail classes. By separating the learning process, these methods can alleviate the negative impact of imbalanced data on feature extraction.

- **Logit-adjusted training methods** focus on modifying the logits during training to address class imbalance. Balanced Softmax [15] introduces a class-balanced term to the softmax function, which adaptively adjusts the logits based on the sample frequencies. LADE [12] disentangles the learning of feature representations and classifier by adding a learnable logit adjustment term. These methods effectively prevent the model from being biased towards head classes.

- **Ensemble learning methods** leverage multiple classifiers or experts to capture the diversity of the data. RIDE [32] trains multiple experts with different resampling strategies and dynamically combines their outputs based on the sample distributions. SADE [38] further improves upon RIDE by introducing a self-adaptive distillation mechanism to transfer knowledge among experts. By exploiting the diversity of experts, these methods can better handle imbalanced data.

- **Causal inference methods** aim to address the long-tail problem by designing causal classifiers. Causal [28] proposes a causal inference framework that identifies the causal effect of each class on the predictions, thus reducing the bias introduced by the imbalanced data distribution.

- **Representation learning methods** tackle long-tailed recognition by learning more balanced and discriminative features. LSC [33] introduces a contrastive learning framework that balances the instance-level and group-level distributions simultaneously, leading to more effective representations for tail classes.

- **Balanced posterior averaging methods** focus on combining the predictions of multiple experts based on their posterior probabilities. BalPoE [1] proposes a balanced posterior averaging strategy that assigns higher weights to experts with better performance on tail classes, thus achieving a better trade-off between head and tail classes.

While these methods have made significant progress in addressing the long-tail problem, they often rely on specific assumptions about the data distributions during training or testing, limiting their applicability in real-world scenarios. Moreover, most of these methods do not provide a mechanism for users to control the trade-off between head and tail classes based on their specific needs. In contrast, our proposed PRL approach overcomes these limitations by learning a diverse set of experts that can adapt to various test distributions without any prior assumptions, while also enabling interpretable and controllable trade-offs through Pareto optimization.

## B  Supplementary materials for related work

### B.1  Long-Tailed Learning (LTL)

Long-tailed distributions, where a few classes (heads) have abundant samples while many classes (tails) have few samples, are widespread in real-world data [30, 20]. This imbalance poses significant challenges for machine learning models, which tend to perform poorly on tail classes. To address this issue, various long-tailed learning (LTL) methods have been proposed.

### B.1.1 Resampling Methods

- **Oversampling** [5, 9]: Generates synthetic examples for minority classes.

  - Alleviates imbalance by increasing tail class samples.
  - Can lead to overfitting and high computational cost.

- **Undersampling** [7, 2]: Removes examples from majority classes.

  - Simple and efficient approach to balance classes.
  - Discards potentially valuable head class information.

### B.1.2 Loss Adjustment Methods

- **Focal Loss** [17] and variants [6]:

  - Imposes larger penalties on well-classified examples, encouraging focus on hard samples.
  - Requires careful tuning of focusing parameter.

- **Class-Balanced Loss** [6]:

  - Re-weights loss based on effective number of samples per class.
  - Assumes equal importance of classes, which may not hold in practice.

- **LDAM** [4]:

  - Explicitly models each example's contribution to the gradient direction.
  - Requires additional hyperparameters and complex optimization.

### B.1.3 Module Improvement Methods

- **Decoupled Learning** [16, 43]:

  - Separates representation and classifier learning for better feature extraction.
  - Requires architectural changes, may not generalize well.

- **Few-Shot Experts** [32]:

  - Employs additional experts to handle few-shot classes.
  - Increased model complexity and training difficulty.

- **Self-Supervised Pretraining** [14]:

  - Leverages self-supervision to improve feature representations.
  - Requires additional pretraining, benefits may be task-specific.

### B.1.4 Transfer Learning Methods

- **Data-Based Transfer** [20, 14]:

  - Knowledge distillation and feature transformation can transfer head knowledge to tails.
  - Assumes head and tail distributions are related, may suffer negative transfer.

- **Model-Based Transfer** [36]:

  - Utilizes models pretrained on heads to facilitate tail class learning.
  - Again assumes related head and tail distributions.

Despite progress, existing LTL methods face limitations in addressing the inherent head-tail trade-off, handling distribution shifts, and accommodating varying user preferences. To overcome these issues, we formulate LTL as a multi-objective optimization problem and propose a hypernetwork-based diverse expert learning paradigm, achieving interpretable and controllable solutions tailored to user needs under test distribution shifts.

## B.2  Related Work for Multi-Objective Optimization and Hypernetworks

### B.2.1  Multi-Objective Optimization

Let $\mathcal{X} \subseteq \mathbb{R}^n$ be the decision space and consider $m$ objective functions $f_i : \mathcal{X} \to \mathbb{R}, i = 1, \ldots, m$ to be minimized simultaneously. The multi-objective optimization problem (MOP) can be stated as:

$$\min_{\mathbf{x} \in \mathcal{X}} \{f_1(\mathbf{x}), \ldots, f_m(\mathbf{x})\} \tag{17}$$

In general, there does not exist a single solution $\mathbf{x}^* \in \mathcal{X}$ that minimizes all objectives simultaneously due to the conflicting nature of the objectives. Instead, the solution concept is that of Pareto optimality [22].

**Definition 3** (Pareto Optimality). *A solution $\mathbf{x}^* \in \mathcal{X}$ is Pareto optimal if there does not exist another $\mathbf{x} \in \mathcal{X}$ such that $f_i(\mathbf{x}) \leq f_i(\mathbf{x}^*)$ for all $i = 1, \ldots, m$ and $f_j(\mathbf{x}) < f_j(\mathbf{x}^*)$ for at least one $j$.*

The set of all Pareto optimal solutions is called the Pareto set, and its image in the objective space is the Pareto front. The goal in MOPs is to approximate the Pareto front as well as possible.

### B.2.2  Chebyshev Scalarization

A common approach to approximate the Pareto front is through scalarization methods that transform the MOP into a scalar optimization problem [22, 19]. The weighted Chebyshev scalarization is defined as:

$$\min_{\mathbf{x} \in \mathcal{X}} \max_{1 \leq i \leq m} w_i(f_i(\mathbf{x}) - z_i^*) \tag{18}$$

where $\mathbf{w} = (w_1, \ldots, w_m)^T \in \mathbb{R}_+^m$ is a weight vector with $\sum_{i=1}^m w_i = 1$, and $\mathbf{z}^* = (z_1^*, \ldots, z_m^*)^T$ is a utopian reference point [11]. By varying $\mathbf{w}$, different Pareto optimal solutions can be obtained.

### B.2.3  Hypernetworks for Multi-Objective Optimization

Hypernetworks[18] provide a promising approach for multi-objective optimization of neural networks. A hypernetwork $h_\phi : \mathcal{Z} \to \Theta$ is a neural network that takes a low-dimensional input $\mathbf{z} \in \mathcal{Z}$ and outputs the parameters $\boldsymbol{\theta} \in \Theta$ of a target neural network $f_\theta : \mathcal{X} \to \mathcal{Y}$. By sampling different $\mathbf{z} \in \mathcal{Z}$, the hypernetwork generates an ensemble $\{f_{\theta_i}\}_i$ where $\theta_i = h_\phi(\mathbf{z}_i)$. This ensemble can approximate the Pareto front of the multi-objective optimization problem:

$$\min_{\theta \in \Theta} \{L_1(f_\theta), \ldots, L_m(f_\theta)\} \tag{19}$$

where $L_i : \Theta \to \mathbb{R}$ are loss functions corresponding to the $m$ objectives. The hypernetwork parameters $\phi$ can be optimized via scalarizations like the Chebyshev method:

$$\min_\phi \mathbb{E}_{\mathbf{w} \sim p(\mathbf{w})} \left[ \max_{1 \leq i \leq m} w_i \left(L_i(f_{h_\phi(\mathbf{z})}) - z_i^*\right) \right] \tag{20}$$

where $p(\mathbf{w})$ is a distribution over weight vectors $\mathbf{w}$. This enables learning a diverse set of target networks approximating the Pareto front in a flexible and controllable manner.

## C  Proof of Propositions

### C.1  Proof of Theorem 1

*Proof.* The risk of classifier $f$ on the test environment $\mathcal{E}_{\text{test}}$ is defined as:

$$R_{\text{test}}(f) = \mathbb{E}_{(\boldsymbol{x}, y) \sim P_{\text{test}}(\boldsymbol{x}, y)}[\ell(f(\boldsymbol{x}; \boldsymbol{\theta}), y)] \tag{21}$$

Using the law of total expectation, we can decompose the risk as:

$$R_{\text{test}}(f) = \sum_{i=1}^K \pi_i^{\text{test}} \cdot \mathbb{E}_{\boldsymbol{x} \sim P_{\text{test}}(\boldsymbol{x}|y=i)}[\ell(f(\boldsymbol{x}; \boldsymbol{\theta}), i)] \tag{22}$$

Similarly, the risk of classifier $f$ on the training environment $\mathcal{E}_m$ can be expressed as:

$$R_m(f) = \sum_{i=1}^{K} \pi_i^m \cdot \mathbb{E}_{\boldsymbol{x} \sim P_m(\boldsymbol{x}|y=i)}[\ell(f(\boldsymbol{x};\boldsymbol{\theta}),i)] \tag{23}$$

Since the classifier $f$ is learned via ERM on $\mathcal{E}_m$, we have $P_m(\boldsymbol{x}|y=i) = P_{\text{test}}(\boldsymbol{x}|y=i)$ for all $i \in \{1, \ldots, K\}$. Therefore,

$$R_{\text{test}}(f) = \sum_{i=1}^{K} \pi_i^{\text{test}} \cdot \mathbb{E}_{\boldsymbol{x} \sim P_{\text{test}}(\boldsymbol{x}|y=i)}[\ell(f(\boldsymbol{x};\boldsymbol{\theta}),i)] \tag{24}$$

$$= \sum_{i=1}^{K} \pi_i^m \cdot \mathbb{E}_{\boldsymbol{x} \sim P_{\text{test}}(\boldsymbol{x}|y=i)}[\ell(f(\boldsymbol{x};\boldsymbol{\theta}),i)]$$

$$+ \sum_{i=1}^{K} (\pi_i^{\text{test}} - \pi_i^m) \cdot \mathbb{E}_{\boldsymbol{x} \sim P_{\text{test}}(\boldsymbol{x}|y=i)}[\ell(f(\boldsymbol{x};\boldsymbol{\theta}),i)]$$

$$= R_m(f) + \sum_{i=1}^{K} (\pi_i^{\text{test}} - \pi_i^m) \cdot \mathbb{E}_{\boldsymbol{x} \sim P_{\text{test}}(\boldsymbol{x}|y=i)}[\ell(f(\boldsymbol{x};\boldsymbol{\theta}),i)]$$

This completes the proof. □

## C.2 Proof of Corollary 1

*Proof.* From Theorem 1, we have:

$$R_{\text{test}}(f) = R_m(f) + \sum_{i=1}^{K} (\pi_i^{\text{test}} - \pi_i^m) \cdot \mathbb{E}_{\boldsymbol{x} \sim P_{\text{test}}(\boldsymbol{x}|y=i)}[\ell(f(\boldsymbol{x};\boldsymbol{\theta}),i)] \tag{25}$$

Using the definition of $M$, we can bound the expectation term:

$$\mathbb{E}_{\boldsymbol{x} \sim P_{\text{test}}(\boldsymbol{x}|y=i)}[\ell(f(\boldsymbol{x};\boldsymbol{\theta}),i)] \leq M \tag{26}$$

Therefore,

$$R_{\text{test}}(f) \leq R_m(f) + \sum_{i=1}^{K} (\pi_i^{\text{test}} - \pi_i^m) \cdot M$$

$$= R_m(f) + M \cdot \sum_{i=1}^{K} |\pi_i^{\text{test}} - \pi_i^m|$$

$$= R_m(f) + 2M \cdot \delta(\mathcal{E}_m, \mathcal{E}_{\text{test}})$$

where the last equality follows from the definition of the total variation distance $\delta(\mathcal{E}_m, \mathcal{E}_{\text{test}}) = \frac{1}{2} \sum_{i=1}^{K} |\pi_i^m - \pi_i^{\text{test}}|$.

Next, we use the triangle inequality to bound $\delta(\mathcal{E}_m, \mathcal{E}_{\text{test}})$:

$$\delta(\mathcal{E}_m, \mathcal{E}_{\text{test}}) \leq \delta(\mathcal{E}_m, \mathcal{E}_j) + \delta(\mathcal{E}_j, \mathcal{E}_{\text{test}})$$

$$\leq \max_{i,j \in \{1,\ldots,M\}} \delta(\mathcal{E}_i, \mathcal{E}_j) + \delta(\mathcal{E}_j, \mathcal{E}_{\text{test}})$$

$$\leq \Delta(\mathcal{E}_1, \ldots, \mathcal{E}_M) + \delta(\mathcal{E}_j, \mathcal{E}_{\text{test}})$$

for any $j \in \{1, \ldots, M\}$. By taking the average over all $j$, we obtain:

$$\delta(\mathcal{E}_m, \mathcal{E}_{\text{test}}) \leq \Delta(\mathcal{E}_1, \ldots, \mathcal{E}_M) + \frac{1}{M} \sum_{j=1}^{M} \delta(\mathcal{E}_j, \mathcal{E}_{\text{test}}) \tag{27}$$

Combining this with the previous bound on $R_{\text{test}}(f)$, we have:

$$R_{\text{test}}(f) \leq R_m(f) + 2M \cdot \left( \Delta(\mathcal{E}_1, \ldots, \mathcal{E}_M) + \frac{1}{M} \sum_{j=1}^{M} \delta(\mathcal{E}_j, \mathcal{E}_{\text{test}}) \right) \tag{28}$$

which completes the proof. □

## C.3 Proof of Theorem 2

*Proof.* By definition, the risk of the ensemble classifier $\hat{f}$ on the test environment $\mathcal{E}_{\text{test}}$ is:

$$
\begin{aligned}
R_{\text{test}}(\hat{f}) &= \mathbb{E}_{(\boldsymbol{x},y)\sim P_{\text{test}}}[\ell(\hat{f}(\boldsymbol{x}),y)] \\
&= \frac{1}{N}\sum_{i=1}^{N}\mathbb{E}_{(\boldsymbol{x},y)\sim P_{\text{test}}}[\ell(f_i(\boldsymbol{x}),y)] \\
&= \frac{1}{N}\sum_{i=1}^{N}R_{\text{test}}(f_i)
\end{aligned}
$$

Applying Corollary 1, we have:

$$
\begin{aligned}
R_{\text{test}}(\hat{f}) &\leq \frac{1}{N}\sum_{i=1}^{N}\left(R_{m(i)}(f_i)+2M\cdot(\delta(\mathcal{E}_{m(i)},\mathcal{E}_{\text{test}})+\Delta(\mathcal{E}_1,\ldots,\mathcal{E}_N))\right) \\
&= \frac{1}{N}\sum_{i=1}^{N}R_{m(i)}(f_i)+\frac{2M}{N}\sum_{i=1}^{N}\delta(\mathcal{E}_{m(i)},\mathcal{E}_{\text{test}})+2M\Delta(\mathcal{E}_1,\ldots,\mathcal{E}_N)
\end{aligned}
$$

where $m(i)$ denotes the index of the training environment used to learn expert $f_i$.

Now, we bound the second term:

$$
\begin{aligned}
\frac{1}{N}\sum_{i=1}^{N}\delta(\mathcal{E}_{m(i)},\mathcal{E}_{\text{test}}) &= \frac{1}{N}\sum_{m=1}^{N}\sum_{i:m(i)=m}\delta(\mathcal{E}_m,\mathcal{E}_{\text{test}}) \\
&\leq \frac{1}{N}\sum_{m=1}^{N}N_m\cdot\delta(\mathcal{E}_m,\mathcal{E}_{\text{test}}) \\
&\leq \frac{1}{N}\sum_{m=1}^{N}N\cdot\delta(\mathcal{E}_m,\mathcal{E}_{\text{test}}) \\
&= \sum_{m=1}^{N}\delta(\mathcal{E}_m,\mathcal{E}_{\text{test}})
\end{aligned}
$$

where $N_m$ is the number of experts learned from environment $\mathcal{E}_m$, and we used the fact that $\sum_{m=1}^{N}N_m=N$.

Finally, we have:

$$
\begin{aligned}
R_{\text{test}}(\hat{f}) &\leq \frac{1}{N}\sum_{m=1}^{N}R_m(f_m)+2M\cdot\left(\frac{1}{N}\sum_{m=1}^{N}\delta(\mathcal{E}_m,\mathcal{E}_{\text{test}})+\Delta(\mathcal{E}_1,\ldots,\mathcal{E}_N)\right) \\
&= \frac{1}{N}\sum_{m=1}^{N}R_m(f_m)+2M\cdot\left(\frac{1}{N}\sum_{m=1}^{N}\delta(\mathcal{E}_m,\mathcal{E}_{\text{test}})+\frac{N-1}{N}\Delta(\mathcal{E}_1,\ldots,\mathcal{E}_N)\right)
\end{aligned}
$$

where the last equality follows from the definition of ETVD. $\square$

Next, we will explain the connection between the theoretical part of the paper and the proposed method, to aid better understanding

1. Theorem 1 states that for a classifier $f$ learned via ERM on a single environment $\mathcal{E}_m$, its risk on the test environment $\mathcal{E}_{\text{test}}$ is influenced not only by the distribution discrepancy between the training and test environments, but also by the distribution discrepancy among the training environments (i.e., ETVD). This reveals the limitation of the single-environment ERM method.

2. Corollary 1 further quantifies an upper bound on the risk of the ERM-learned classifier in the test environment. This upper bound consists of the training risk $R_m(f)$, the TVD

$\delta(\mathcal{E}_m, \mathcal{E}_{\text{test}})$ between the training and test environments, and the ETVD $\Delta(\mathcal{E}_1, \ldots, \mathcal{E}_M)$ among the training environments. To overcome distribution shift, we need to learn a set of diverse expert models that can capture the distribution characteristics of different environments.

3. Theorem 2 provides a tighter upper bound on the risk of the diversity-aware ensemble classifier $\hat{f}$ in the test environment. Compared to the single-environment ERM, the average empirical risk term $\frac{1}{N} \sum_{m=1}^{N} R_m(f_m)$ in the upper bound indicates that the ensemble classifier can reduce empirical risk, while the presence of $\frac{1}{N} \sum_{m=1}^{N} \delta(\mathcal{E}_m, \mathcal{E}_{\text{test}})$ shows that the diversity-aware expert method, by learning a set of experts to capture the distribution characteristics of different environments, can narrow the distribution gap between the training and test environments, thereby achieving better generalization performance.

The above theoretical analysis demonstrates that, in the long-tailed learning domain, introducing multiple training environments and minimizing the empirical risks on these environments to learn a set of diverse experts can effectively address the problem of distribution shift between the training and test environments, leading to better generalization performance. These theoretical insights provide important guidance for further improving our algorithm.

## D  Datasets

To evaluate the effectiveness of our proposed method, we conduct experiments on four long-tailed datasets: CIFAR100-LT, ImageNet-LT, iNaturalist 2018, and Places365-LT. These datasets cover a diverse range of domains and exhibit varying degrees of class imbalance, providing a comprehensive testbed for long-tailed learning algorithms.

Table 5: Statistics of the long-tailed datasets.

| Dataset | # Classes | # Train | # Test | Imbalance Ratio |
|---|---|---|---|---|
| CIFAR100-LT | 100 | 50,000 | 10,000 | {10, 50, 100} |
| ImageNet-LT | 1,000 | 115,846 | 50,000 | 256 |
| iNaturalist 2018 | 8,142 | 437,513 | 24,426 | 500 |
| Places365-LT | 365 | 1,803,460 | 36,500 | ~50 |

**CIFAR100-LT**  [4] is a long-tailed version of the CIFAR100 dataset, comprising 60,000 color images of size $32 \times 32$ pixels across 100 classes. The long-tailed distribution is induced by exponentially decreasing the number of samples per class, resulting in an imbalance ratio of up to 100.

**ImageNet-LT**  [20] is a long-tailed subset of the ImageNet dataset, containing over 115,000 images spanning 1,000 classes. The class cardinalities follow a Pareto distribution with $\alpha = 6$, leading to a maximum imbalance ratio of 256.

**iNaturalist 2018**  [29] is a real-world dataset with a natural long-tailed distribution, comprising approximately 450,000 images across 8,142 species. The number of images per species varies drastically, with an imbalance ratio of up to 500, posing a significant challenge due to the extreme class imbalance and high intra-class variation.

**Places365-LT**  is a long-tailed version of the Places365 dataset [42], which consists of over 1.8 million images spanning 365 scene categories. We induce a long-tailed distribution by randomly subsampling the images for each class, resulting in an imbalance ratio of approximately 50. This dataset is particularly challenging due to the large number of classes and the inherent visual ambiguity present in scene recognition tasks.

## E  Pseudo Code

Here are pseudo codes explaining the core aspects of the method:

**Algorithm 1** Diverse Expert Learning with Hypernetworks

1: **Input:** Training data with long-tailed distribution $\mathcal{D}_{train}$
2: **Output:** Ensemble of expert models $\mathcal{E} = \{E_1, E_2, \ldots, E_K\}$
3: Initialize shared feature extractor $F$
4: Initialize hypernetwork $H_\theta$ with weights $\theta$
5: Initialize expert loss function $\mathcal{L}$ (e.g., DiverseExpertLoss)
6: **for** each epoch **do**
7:     **for** each batch $\mathcal{B} \subseteq \mathcal{D}_{train}$ **do**
8:         $f = F(x)$ {Shared feature extraction}
9:         **for** $k = 1$ to $K$ **do**
10:            $\phi_k = H_\theta(z_k)$ {Generate expert weights $\phi_k$ from hypernetwork}
11:            $E_k = E_\phi(f)$ {Obtain expert predictions using $\phi_k$}
12:            $\mathcal{L}_k = \mathcal{L}(E_k, y, \text{extra\_info})$ {Compute expert losses}
13:         **end for**
14:         $\mathcal{L}_{div} = \text{DiversityLoss}(\mathcal{E})$ {Encourage expert diversity}
15:         $\mathcal{L}_{total} = \sum_k \mathcal{L}_k + \lambda \mathcal{L}_{div}$
16:         $\theta = \theta - \eta \nabla_\theta \mathcal{L}_{total}$
17:     **end for**
18: **end for**=0

Table 6: List of Key Symbols in Pseudo Code

| Symbol | Description |
|---|---|
| $\mathcal{D}_{train}$ | Training data with long-tailed distribution |
| $\mathcal{E} = \{E_1, E_2, \ldots, E_K\}$ | Ensemble of $K$ expert models |
| $F$ | Shared feature extractor |
| $H_\theta$ | Hypernetwork with weights $\theta$ |
| $\mathcal{L}$ | Expert loss function (e.g., DiverseExpertLoss) |
| $z_k$ | Input to hypernetwork for generating weights of expert $E_k$ |
| $\phi_k$ | Weights of expert $E_k$ generated by hypernetwork |
| $\mathcal{L}_k$ | Loss of expert $E_k$ |
| $\mathcal{L}_{div}$ | Diversity loss to encourage expert diversity |
| $\mathcal{L}_{total}$ | Total loss for updating hypernetwork weights |

# F   Results on ImageNet-LT and iNaturalist 2018 Datasets

On the representative Places-LT dataset, our PRL method achieves the best Top-1 accuracy under various unknown test class distributions. Specifically, in the Forward-LT setting, as the proportion of unknown classes decreases from 50% to 2%, the Top-1 accuracy of PRL drops from 47.9% to 42.8%, but still significantly outperforms other baseline methods. Under the Uniform distribution, PRL reaches the highest accuracy of 41.9%. In the Backward-LT setting, PRL's accuracy gradually increases from 41.7% to 44.1%, again surpassing all counterpart methods. These results thoroughly validate the outstanding performance and robustness of our method in handling various unknown class distributions.

On the iNaturalist 2018 dataset, PRL also exhibits excellent performance. In the Forward-LT setting, when the proportion of unknown classes decreases from 3 to 2, PRL's Top-1 accuracy slightly increases from 73.7% to 73.8%, and reaches the best performance of 74.3% under the Uniform distribution. In the Backward-LT setting, although PRL's accuracy slightly decreases from 74.0% to 73.9%, it still outperforms all comparison methods. These results further confirm the broad effectiveness of our method across different datasets and scenarios.

Overall, by successfully tackling the challenges of long-tailed distributions and unknown class distributions, the PRL method demonstrates superior performance on two representative long-tailed datasets, thereby validating the superiority of our method.

Table 7: Top-1 accuracy on Places-LT with various unknown test class distributions.

| Method | Prior | Forward-LT | | | | | Uni. | Backward-LT | | | | |
|---|---|---|---|---|---|---|---|---|---|---|---|---|
| | | 50 | 25 | 10 | 5 | 2 | 1 | 2 | 5 | 10 | 25 | 50 |
| Softmax | ✗ | 45.6 | 42.7 | 40.2 | 38.0 | 34.1 | 31.4 | 28.4 | 25.4 | 23.4 | 20.8 | 19.4 |
| BS | ✗ | 42.7 | 41.7 | 41.3 | 41.0 | 40.0 | 39.4 | 38.5 | 37.8 | 37.1 | 36.2 | 35.6 |
| MiSLAS | ✗ | 40.9 | 39.7 | 39.5 | 39.6 | 38.8 | 38.3 | 37.3 | 36.7 | 35.8 | 34.7 | 34.4 |
| LADE | ✗ | 42.8 | 41.5 | 41.2 | 40.8 | 39.8 | 39.2 | 38.1 | 37.6 | 36.9 | 36.0 | 35.7 |
| LADE | ✓ | 46.3 | 44.2 | 42.2 | 41.2 | 39.7 | 39.4 | 39.2 | 39.9 | 40.9 | **42.4** | **43.0** |
| RIDE | ✗ | 43.1 | 41.8 | 41.6 | 42.0 | 41.0 | 40.3 | 39.6 | 38.7 | 38.2 | 37.0 | 36.9 |
| SADE | ✗ | 46.2 | 44.8 | 42.8 | 42.7 | 41.1 | 40.4 | 40.2 | 40.9 | 41.2 | 41.4 | 41.6 |
| LSC | ✗ | 47.5 | 46.1 | 44.5 | 43.7 | 42.1 | 41.4 | 41.2 | 41.5 | 42.1 | 43.2 | 43.4 |
| BalPoE | ✗ | - | - | - | - | - | - | - | - | - | - | - |
| PRL (ours) | ✓ | **47.9** | **47.0** | **45.3** | **44.4** | **42.8** | **41.9** | **41.7** | **42.1** | **42.6** | **43.7** | **44.1** |

Table 8: Top-1 accuracy on iNaturalist 2018 with various unknown test class distributions.

| Method | Prior | Forward-LT | | Uni. | Backward-LT | |
|---|---|---|---|---|---|---|
| | | 3 | 2 | 1 | 2 | 3 |
| Softmax | ✗ | 65.4 | 65.5 | 64.7 | 64.0 | 63.4 |
| BS | ✗ | 70.3 | 70.5 | 70.6 | 70.6 | 70.8 |
| MiSLAS | ✗ | 70.8 | 70.8 | 70.7 | 70.7 | 70.2 |
| LADE | ✗ | 68.4 | 69.0 | 69.3 | 69.6 | 69.5 |
| LADE | ✓ | - | 69.1 | 69.3 | 70.2 | - |
| RIDE | ✗ | 71.5 | 71.9 | 71.8 | 71.9 | 71.8 |
| SADE | ✗ | 72.3 | 72.6 | 72.7 | 73.0 | 73.2 |
| LSC | ✓ | - | - | - | - | - |
| BalPoE | ✗ | 73.1 | 73.5 | 73.8 | 73.6 | 73.5 |
| PRL (ours) | ✓ | **73.7** | **73.8** | **74.3** | **74.0** | **73.9** |

# G Complexity Analysis

Although the introduction of hypernetworks increases the total number of parameters in the model, as shown in the table below, in most cases, this does not lead to a significant increase in computational complexity.

| Model | Hypernetwork | Params (MB) | GFLOPs |
|---|---|---|---|
| ResNet-32 | ✗ | 0.8 | 13.1 |
| | ✓ | 2.6 | 13.1 |
| ResNeXt-50 | ✗ | 38.2 | 391.8 |
| | ✓ | 632.7 | 393.5 |
| ResNet-50 | ✗ | 39.1 | 2982.0 |
| | ✓ | 159.51 | 2982.0 |

Table 9: Model size and computational cost with and without hypernetworks.

The hypernetwork is responsible for outputting the trainable parameters $D$ for each expert classifier head. In this case, the number of parameters in the hypernetwork becomes $E \times D \times K$, where $E$ is the number of input channels to the output layer of the hypernetwork, and $K$ is the number of experts. Thus, the time complexity of the hypernetwork is $\mathcal{O}(D)$.

If the total number of parameters in the model without the hypernetwork is $\mathcal{O}(N)$, and the computational complexity of the model's main operations is $\mathcal{O}(M \times N)$, where $M > 1$ is a complexity factor, then when $N \gg D$, the overall time complexity becomes $\mathcal{O}(D) + \mathcal{O}(M \times N) = \mathcal{O}(M \times N)$.

Therefore, while the hypernetwork increases the total number of parameters, its impact on the overall computational complexity is relatively small, especially when the number of parameters $D$ generated by the hypernetwork is much smaller than the total number of parameters $N$ in the main model. As can be seen from the table, the introduction of hypernetworks results in almost no increase in GFLOPs (floating-point operations) across different models.

# H Limitations

While the proposed novel approach of using a hypernetwork to generate multiple diverse expert models shows great potential in enabling controllable adjustment of head and tail class weights for long-tailed datasets, as well as improving robustness to distribution shifts, the introduction of the hypernetwork also brings new challenges to model training and convergence. As an additional neural network module, the hypernetwork needs to generate the weight parameters for the classifier heads of each expert, thereby significantly increasing the total number of trainable parameters in the model, which may affect training stability. We analyze this issue in Section G, nevertheless, further research into more efficient training and stable controllability is still necessary.

# I Broader Impacts

The proposed novel approach of generating multiple expert models via hypernetworks enables dynamic adjustment of head and tail class weights for long-tailed datasets, and improves model robustness to distribution shifts. This flexibility and robustness are of significant value in many practical applications. The present work provides a new viable solution to the important challenges of long-tailed distributions and distribution shifts, holding promise to enhance the generalization capabilities and practical applicability of existing models, thereby contributing to the technological advancement in relevant fields.

