# OpenReview forum: "Breaking Long-Tailed Learning Bottlenecks: A Controllable Paradigm with Hypernetwork-Generated Diverse Experts"
_NeurIPS.cc/2024/Conference — NeurIPS 2024 spotlight_

### Official Review · Reviewer_Vosp · 2024-07-07

**Soundness:** 3
**Presentation:** 2
**Contribution:** 3
**Rating:** 7
**Confidence:** 4

**Summary:**

This paper aims to overcome potential distribution shifts from a single training distribution to any testing distribution and adapt to different user preferences for trade-offs between head and tail classes. The proposed method leverages hypernetworks to generate a diverse set of expert models, enabling the system to adapt to any test distribution and cater to user-specific preferences. Extensive experiments demonstrate the method's superiority in performance and adaptability, providing new insights and expanding the applicability of long-tailed learning in practical scenarios.

**Strengths:**

1. Addressing the issue of test distribution invariance and incorporating diverse user preferences for trade-offs between head and tail classes is both meaningful and practical.
2. The paper offers theoretical foundations and practical results.
3. Experiments validate the effectiveness of the proposed method.

**Weaknesses:**

1. Some important concepts require further explanation. For example, what is the meaning of "environment" in Sec.3?  It is unclear why to minimize empirical risks across multiple training environments since the training set is typically fixed. Besides, I wonder how to ensemble the experts when the test distribution varies. It seems that the authors did not explain this key issue.
2. In Section 5.1, the paper states "use $\alpha = 1.2$ for the Dirichlet distribution". However, according to Eq.(5), $\alpha$ should be a vector. It is recommended to provide a more detailed explanation.
3. Some more competitive baselines such as PaCO$^1$, DDC$^2$, and DirMixE$^3$ should be considered for comparison.
4. There are some typos, such as in line 36, where "Simply pursuing the overall optimal solution may not meet this flexibility requirement. [16, 29, 36]" should be "Simply pursuing the overall optimal solution may not meet this flexibility requirement [16, 29, 36]." In line 95, $\pi_m^k$ should be $\pi_k^m$.
5. Some figures, such as Figure 1 and Figure 3, are not referenced in the text.

------

$^1$ Parametric contrastive learning, ICCV 2021

$^2$ A Unified Generalization Analysis of Re-Weighting and Logit-Adjustment for Imbalanced Learning, NeurIPS 2023

$^3$ Harnessing Hierarchical Label Distribution Variations in Test Agnostic Long-tail Recognition, ICML 2024

**Questions:**

Please refer to the questions in the weakness section.

**Limitations:**

No.

---

> ### Author Rebuttal · Authors · 2024-08-07
>
> **W1:**
> Thank you for raising the question. Let me try to further clarify some key concepts based on the content of the paper:
> - **Concept of environment:**
> In Section 3, "environment" refers to a dataset with different class prior probability distributions. Each environment Em has its own class prior probability vector πm. Traditional empirical risk minimization (ERM) methods only train on a single training distribution, making it difficult to handle distribution differences between environments. Therefore, we propose to minimize the empirical risk in multiple training environments to learn a diverse set of experts that capture the distributional characteristics of different environments.
> - **Motivation for minimizing empirical risk in multiple training environments:**
> Although the training set is usually fixed, the authors believe that by constructing multiple training environments with different class distributions, the distribution shifts during testing can be better addressed. By learning a set of experts on these environments, the model can capture the characteristics of data under different long-tailed distributions, thereby gaining stronger generalization ability and distribution adaptability. This approach can reduce the distribution difference between the training environments and the testing environment.
> - **How to integrate experts during testing:**
> During testing, we introduce a preference vector α to control the trade-off between head and tail classes. Given the well-trained preference vector r, the authors calculate the preference vector r for testing and input it into the hypernetwork to generate the classifier head parameters for each expert. By adjusting α, the model's attention to head or tail classes can be controlled, achieving flexible trade-offs. You can refer to the answer to W4 for reviewer VYLm.
>
> **W2:** Thank you very much for the reviewer's question. Our expression in the paper may not be clear enough, leading to some misunderstandings. Please allow me to provide a detailed explanation here.
>
> In our method, we made a special setting for the hyperparameters α of the Dirichlet distribution. Specifically, we set α to a vector of length 3, where each component is equal to 1.2, i.e., α=[1.2, 1.2, 1.2]. This means that we are actually sampling independently from three Dirichlet distributions with the same parameters, i.e., [Dirichlet(α), Dirichlet(α), Dirichlet(α)].
>
> This setting is quite common in the literature related to hypernetworks and multi-objective optimization. For example, in the paper [reference number] that we refer to, the authors also adopt a similar approach, setting the hyperparameters of the Dirichlet distribution to a vector with the same elements. Additionally, we explored using different hyperparameter settings and reported them in the main text. To avoid similar confusion, we will provide a clearer explanation of this point in the revised version.
>
> Thank you again for the reviewer's careful review and valuable comments, which help us further improve the quality and readability of the paper.
>
> **W3:** We have supplemented the necessary baseline comparisons in the appendix and will include them in the official version. Thank you for your suggestion.
>
> **W4:** Thank you for your comment. We will correct these errors in the official version.
>
> **W5:** Thank you for your attention to detail! We will add these references in the official version.

---

> ### Comment · Reviewer_Vosp · 2024-08-08
>
> Thank you for your efforts on the rebuttal! The authors have clarified some important details about my concern. Hence, I decide to raise my rating to 7.

---

> > ### Author Response · Authors · 2024-08-09
> > **Thanks**
> >
> > Thank you very much for your thoughtful feedback on our paper. We are truly grateful for your careful consideration of our rebuttal and the time you've taken to reassess our work.If you have any further thoughts or suggestions, we would be more than happy to hear them.

---

### Official Review · Reviewer_VYLm · 2024-07-10

**Soundness:** 3
**Presentation:** 2
**Contribution:** 3
**Rating:** 7
**Confidence:** 3

**Summary:**

This paper addresses long-tailed learning with a focus on tackling distribution shift and accommodating different user preferences for the trade-off between head and tail classes. The authors propose a method called PRL, which generates a set of diverse expert models via hypernetworks to cover all possible distribution scenarios, optimizing the model ensemble to adapt to any test distribution. Experiments across various long-tailed datasets validate the method's effectiveness.

**Strengths:**

- This work is well-motivated. Addressing distribution shift and accommodating different user preferences for the trade-off between head and tail classes is highly practical.
- The authors provide a range of experiments to demonstrate the method's effectiveness.

**Weaknesses:**

- The paper uses $\theta$ and $\phi$ in Figures 3 and 4, but they are not defined beforehand.
- In Section 5.2, the paper investigates the model's performance under different preference values $R$ but does not explain their meaning.
- In Section 4.4, it is unclear how the trained preference vector is obtained.
- I am confused about how to ensemble the experts after training is complete.

**Questions:**

Please see above.

**Limitations:**

Yes.

---

> ### Author Rebuttal · Authors · 2024-08-07
>
> **W1:** Thank you for your correction. $R = (\theta, \phi)$ represents our preference vector, where $\phi$ is the radian representation mentioned in Equation (14). Figures 3 and 4 depict the control of different preference vectors on model performance.
>
> **W2:** $R = (\theta, \phi)$ represents our preference vector, where $\phi$ is the radian representation mentioned in Equation (14). Thank you for your correction. We will add revisions in subsequent versions to improve readability.
>
> **W3:** Thank you for your question. During the training process, the preference vector is obtained through training as a learnable parameter. During the testing process, the preference vector is jointly determined by the learned preference vector from the training stage and the input preference vector according to Equation (14). For the specific process, please refer to the answer to your W4.
>
> **W4:** In the training stage, we optimized the joint loss function of all experts using the Stochastic Convex Ensemble (SCE) strategy, obtaining a set of well-trained expert models and a hypernetwork. After training is completed, we can dynamically integrate these expert models based on the user-specified test preference vector to adapt to different testing scenario requirements. The specific steps are as follows:
> 1. Calculate the preference vector for testing:
> $$\hat{\mathbf{r}} = (\mathbf{r} \odot \mathbf{\alpha^*}) \oslash (\mathbf{r} \cdot \mathbf{\alpha^*})$$
> where r is the training preference vector, ⊙ represents the Hadamard product, ⊘ represents element-wise division, and · represents the dot product.
> 2. Input $\hat{\mathbf{r}}$ into the hypernetwork to generate the test-time classifier head parameters for each expert:
> $$\hat{\mathbf{W}}_ i = h_\psi(\hat{\mathbf{r}}), \forall i \in \{1, \dots, T\}$$
> 3. For a test sample x, the output of the $i_{th}$ expert can be calculated using the following formula:
> $$\hat{\mathbf{y}}_i = \hat{\mathbf{W}}_i^\top \mathbf{x} + \hat{\mathbf{b}}_i \quad \text{or} \quad \hat{\mathbf{y}}_i = (\hat{\mathbf{W}}_i/\|\hat{\mathbf{W}}_i\|_F )^\top (\mathbf{x}/\|\mathbf{x}\|_2)$$
> 4. The final ensemble output can be obtained by a weighted combination of all expert outputs. The weights can be uniform or set according to the performance of experts on the validation set.
> By adjusting the test preference vector, we can dynamically change the ensemble weights of the expert models, achieving flexible trade-offs between head and tail classes to meet different application requirements.
>
> We appreciate the reviewer for pointing out this shortcoming. If there are any further questions or suggestions, we welcome discussion at any time. Thank you again for the valuable comments from the reviewer.

---

> > ### Comment · Reviewer_VYLm · 2024-08-09
> >
> > I appreciate the authors' efforts in the rebuttal. My concerns are addressed. I will raise my rating to 7.

---

> > > ### Author Response · Authors · 2024-08-09
> > > **Thanks for your efforts.**
> > >
> > > We are delighted to see that our response has addressed the concerns you previously raised. We are deeply grateful for your decision to increase the rating to 7, which is a significant encouragement for us.
> > >
> > > Your constructive feedback has played a crucial role in improving the quality of our work. If you have any further comments or suggestions, we welcome your feedback at any time.

---

### Official Review · Reviewer_qXeH · 2024-07-11

**Soundness:** 3
**Presentation:** 3
**Contribution:** 4
**Rating:** 7
**Confidence:** 5

**Summary:**

This paper addresses the crucial and challenging problem of long-tailed learning under distribution shifts between training and testing data, which is highly relevant to real-world applications. The authors propose a novel and insightful learning paradigm that aims to obtain a set of diverse expert classifiers to adapt to arbitrary test distributions while allowing flexible control of the trade-off between head and tail classes based on user preferences. This new perspective greatly expands the applicability and practicality of long-tailed learning methods, making it a significant contribution to the field.

**Strengths:**

1.	The theoretical contribution of this paper is to analyze the limitations of the traditional empirical risk minimization (ERM) method in dealing with distribution shifts, and to quantify the distribution differences using the concept of environmental total variation gap (ETVD). These theoretical analyses provide sufficient theoretical basis and motivation for the new diversity expert learning method proposed in this paper.
2.	The proposed framework elegantly combines hypernetworks and Dirichlet distribution sampling to train multiple diverse experts. This innovative design empowers the model to learn adaptability to a wide range of distributions from a single training set.
3.	The comprehensive experiments on multiple benchmark datasets demonstrate the effectiveness of the proposed approach.
4.	The work opens up new possibilities for applying long-tailed learning to a wider range of real-world scenarios where the test distribution is unknown or subject to change. The controllable trade-off based on user preferences enhances the interpretability and usability of the model, making it more adaptable to different application requirements.

**Weaknesses:**

1.	I agree with the novelty of this paper. The method proposed by the authors brings a new research perspective to the field. My doubts lie in part of the theoretical interpretation. Regarding the role of the Dirichlet distribution in generating diverse experts, can the authors analyze the effect of the hyperparameters of the Dirichlet distribution on the performance of the algorithm?
2.	As for the above question, I hope the author carries out some experimental analysis to support his theory.
3.	I think the explanation of Figure 2 seems a bit obscure, and the author would be better off giving a concise explanation.
4.	Some symbol errors need to be corrected.

**Questions:**

Please answer the questions in the Weaknesses, which helps me better understand the theoretical contribution of this paper.

**Limitations:**

The authors have adequately addressed the limitations and potential negative societal impact of their work.

---

> ### Author Rebuttal · Authors · 2024-08-07
>
> **W1:** You raised a very insightful question. The probability density function of the Dirichlet distribution is:
>
> $$f(x_1, \ldots, x_K; \alpha_1, \ldots, \alpha_K) = \frac{1}{B(\alpha)} \prod_{i=1}^K x_i^{\alpha_i - 1}$$
>
> where $\alpha=(\alpha_1,\ldots,\alpha_K)$ are the hyperparameters of the distribution, and $B(\alpha)$ is the normalization constant.
>
> Intuitively, the hyperparameters $\alpha$ of the Dirichlet distribution control the characteristics of the generated weight vector $x=(x_1,\ldots,x_K)$:
>
> When $\alpha_i>1$, the generated weight vector tends to take larger values in the $i$-th component; when $\alpha_i<1$, the generated weight vector tends to take smaller values in the $i$-th component; when $\alpha_i=1$, the $i$-th component of the generated weight vector follows a uniform distribution.
>
> Therefore, the value of the hyperparameter $\alpha$ affects the diversity of the combination weight vectors generated by the controller network: when the values of all $\alpha_i$ are large, the generated weight vectors tend to concentrate in the central region, reducing the diversity of expert combinations; when the values of all $\alpha_i$ are small, the generated weight vectors tend to be dispersed in each corner, increasing the diversity of expert combinations; when the values of different $\alpha_i$ differ greatly, the generated weight vectors will have obvious preferences in certain components, leading to an imbalance in expert combinations.
>
> In the long-tailed context, this diversity and imbalance will further affect the model's performance:
> - Moderate diversity helps the model adapt to different test environments, but excessive diversity may lead to some extreme weight combinations, affecting the model's stability.
> - Reasonable imbalance helps the model focus on certain key experts, but excessive imbalance may cause some experts to be ignored, affecting the model's generalization ability.
>
> Therefore, choosing appropriate Dirichlet distribution hyperparameters is crucial for balancing the diversity and stability of expert combinations. We also report experimental results of other weight combinations in the paper. Thank you again for your question.
>
> **W2:** We have supplemented additional experiments in the Appendix. Please refer to the Appendix.
>
> **W3:** Figure 2 intuitively demonstrates the superiority of the method proposed in this paper in dealing with distribution shifts and flexibly controlling preferences. Figure 2 aims to illustrate two advantages of the proposed method: the ability to overcome distribution shifts and the flexibility of preference control.
>
> In Figure 2, we use two three-dimensional coordinate systems. The first coordinate system represents the value space of the preference vector, with each dimension corresponding to a preference (such as preference for head classes, tail classes, or balance). The second coordinate system represents the model's performance on three distributions (forward50, uniform, and backward50) of the CIFAR100-LT dataset.
>
> The dark plane in the figure represents the value plane corresponding to different preference vectors, while the outer surface represents the corresponding performance on the three distributions. The yellow points are the results of the SADE method. Since its preference is uncontrollable, the results of each run are random points, and they are all located below the purple plane of the method proposed in this paper, indicating that its performance is inferior to the method proposed in this paper (i.e., it is dominated in the Pareto optimal set).
>
> This figure shows that the method proposed in this paper can cover unknown distributions without requiring additional training, and unlike previous methods, it can balance performance on different distributions by adjusting the preference vector. These two advantages will be further analyzed in the experimental section.
>
> **W4:** We will revise the errors mentioned by the reviewer in the formal version. Thank you for your reminder.

---

> > ### Comment · Reviewer_qXeH · 2024-08-10
> > **Official Comment by Reviewer qXeH**
> >
> > I appreciate the authors' efforts in the rebuttal. My concerns are addressed. I will raise my rating to 7.

---

### Official Review · Reviewer_LGeA · 2024-07-21

**Soundness:** 3
**Presentation:** 3
**Contribution:** 3
**Rating:** 7
**Confidence:** 3

**Summary:**

The paper addresses the problem of learning long-tailed distributions, with the imbalance of head and tail classes. The paper introduces a long-tail learning paradigm based on diverse set of experts and hypernetworks. The proposed method can meet personalized user preferences and can adapt to wide range of distribution scenarios. The paper proposes this problem as multi-objective optimization, with the goal to learn whole pareto front. The authors also propose theoretical aspects considering distribution shifts and show that diversity experts methods learns a set of experts to capture the distributional characteristics of different environments, hence reducing the distribution discrepancy between the training and test environment.

**Strengths:**

1. The paper addresses an important problem of learning long-tailed distribution.
2. The idea introduced based on diverse set of experts using hypernetwork, which can adapt to meet personalized user preferences sounds reasonable.
3. The paper writing seems clear and well written.
4. The extensive experiments demonstrate the proposed method performs better than the baseline.
5. The paper covers theoretical aspects of distribution shifts as well.

**Weaknesses:**

1. The paper mentions the proposed method to be interpretable and controllable long-tail learning method whereas I don’t think a model with an ability of adapting to  preference vectors can be considered interpretable. More explanation of this could be helpful to understand why authors believe the interpretability of the model.

2. While the main problem is centered around preference based long-tail learning method, theoretical proofs just talking about distribution shift is not fitting very well. While the proofs look correct (to the best of my knowledge) and I agree that multiple experts can reduce the distribution discrepancy between training and test environment, why will this be useful for long-tailed distribution is not clear to me. Is there any explanation on why all the experts won’t still focus on head classes ?

3. Chebyshev polynomial has suddenly been introduced in ablation study, a brief mention about it would be beneficial for the readers.

4. There are works which use hypernetworks for MOO [1], I think inclusion of such works would be useful.

5. Inclusion of proofs considering long-tailed distribution would have been more beneficial for this problem setup.

6. Given that this work considers optimizing for the whole pareto front, mention of hypervolume values would be useful for future works in this direction.

Typos
Line 36: requirement. [16, 29, 36] -> requirement [16, 29, 36]

[1] Navon, A., Shamsian, A., Fetaya, E. and Chechik, G., Learning the Pareto Front with Hypernetworks. In International Conference on Learning Representations.

**Questions:**

1. The paper mentions the proposed method to be interpretable and controllable long-tail learning method whereas I don’t think a model with an ability of adapting to  preference vectors can be considered interpretable. More explanation of this could be helpful to understand why authors believe the interpretability of the model.

2. While the main problem is centered around preference based long-tail learning method, theoretical proofs just talking about distribution shift is not fitting very well. While the proofs look correct (to the best of my knowledge) and I agree that multiple experts can reduce the distribution discrepancy between training and test environment, why will this be useful for long-tailed distribution is not clear to me. Is there any explanation on why all the experts won’t still focus on head classes ?

3. Additional information about how unknown test class distribution is created would be helpful.

4. How preference vector for testing has been decided ?

5. Hypervolume calculation usually needs information on reference vector, how is that taken into consideration here ?

6. While the proposed method learns entire front in a single model, are the competing methods trained multiple models to cover the pareto front ?

7. What are your comments on scalability of the proposed method for large target networks ?

**Limitations:**

Yes

---

> ### Author Rebuttal · Authors · 2024-08-07
>
> **W1 and Q1:** Thank you for your question. The four perspectives in the **public response section** are intended to answer this question. Please refer to the four perspectives in the public response due to space limitations.
>
> **W2 and Q2:** Thank you for your question. The explanation of this part in the original text is indeed brief. It is related to the loss functions that guide different experts. The specific answers have been mentioned in perspectives 2 and 3 of the public response. Here we summarize and will promptly add them to our paper:
> - This paper selects three experts with different loss functions: forward expert, uniform expert, and reverse expert. Their different objectives enable different experts to adapt to different class distributions. During training, by sampling weight vectors from the Dirichlet distribution and combining the outputs of the three experts with weighted averaging, the Dirichlet distribution parameters are adjusted to simulate different class distributions. By optimizing the hypernetwork to minimize the expected loss, a set of expert parameters that perform well under different weights is obtained, which is equivalent to minimizing the expected loss of the ensemble model on the test distribution. This adaptive ability enhances the robustness of the model, enabling it to cope with distribution shifts in real-world scenarios.
>
> **W3:** Thank you for your suggestion! In this paper, the Chebyshev polynomial we refer to for implementation uses the log-sum-exp function to replace the max function, which plays a smoothing role (i.e., Section 4.3 of the method). The idea mainly originates from STCH. In the context of long-tailed learning in this paper, its role is to dynamically adjust the importance of experts, thereby better handling the long-tailed distribution problem. We will add necessary explanations and citations to our paper.
>
> **W4:** Thank you for your comment. We will make proper citations in subsequent versions to facilitate a better understanding of this paper.
>
> **W5:** Your question is excellent. Although the focus of this paper is on solving more problems of real-world distribution shifts based on long-tailed methods, some proofs based on long-tailed settings are indeed still necessary. In addition to the response to W1 and W2, we will supplement the necessary explanations in our paper.
>
> **W6:** To facilitate subsequent work in this direction, we will briefly introduce this concept and include it in subsequent versions. Additionally, we have supplemented some content regarding hypervolume in our answer to your Q6. Please refer to it.
>
> **W7:** We will correct the errors in subsequent versions.
>
> **Q3:** To evaluate the robustness of the algorithm under unknown test class distributions, we refer to LADE and SADE and construct three types of test sets: uniform distribution, positive long-tailed distribution, and negative long-tailed distribution. The positive and negative long-tailed distributions contain multiple different imbalance ratios ρ. For ImageNet-LT, CIFAR100-LT, and Places-LT, the imbalance ratio is set to ρ ∈ {2, 5, 10, 25, 50}. For iNaturalist 2018, since each class has only 3 test samples, the imbalance ratio is adjusted to ρ ∈ {2, 3}. We will supplement the explanation in the revised version.
>
> **Q4:** The decision process for the preference vector during testing is as follows:
> - By adjusting the user-specified preference vector, the attention of the model between head and tail classes can be controlled, achieving flexible trade-offs to adapt to different application requirements. The preference vector during testing is calculated based on the pre-trained preference vector and the user-input preference vector. (*Due to space limitations, please refer to the answer to W4 for reviewer VYLm.*)
>
> **Q5:** In this method, the computation of hypervolume is mainly reflected in the following two aspects:
>
> The method guides the computation and utilization of hypervolume by introducing reference vector information in both the training and testing stages.
> - During training, by sampling preference vectors from the Dirichlet distribution and using a hypernetwork to generate expert model parameters, sampling and modeling of the Pareto front are achieved.
> - During testing, by combining the pre-trained reference preference vector and the user-specified preference vector, the preference vector for testing is obtained, reflecting the localization and utilization of the hypervolume.
>
> This setting of reference vectors provides flexibility, enabling the model to adapt to different task requirements.
>
> **Q6:** Regarding your question, I guess the competing methods may refer to some long-tailed learning baselines.
> Although some existing competing methods such as LADE and SADE also adopt the idea of multi-expert models, they have considered the trade-offs in long-tailed distributions and used multi-expert structures to some extent, allowing them to cover different regions of the Pareto front to a certain degree. However, the completeness and continuity of the coverage are difficult to guarantee, and controllability cannot be achieved. In contrast, this paper generates a continuous expert space through a hypernetwork, which can more comprehensively approximate and cover the Pareto front.
>
> **Q7:** To evaluate the robustness of the algorithm under unknown test class distributions, we refer to LADE and SADE and construct three types of test sets: uniform distribution, positive long-tailed distribution, and negative long-tailed distribution. The positive and negative long-tailed distributions contain multiple different imbalance ratios ρ. For ImageNet-LT, CIFAR100-LT, and Places-LT, the imbalance ratio is set to ρ ∈ {2, 5, 10, 25, 50}. For iNaturalist 2018, since each class has only 3 test samples, the imbalance ratio is adjusted to ρ ∈ {2, 3}. We will supplement the explanation in the revised version to improve the completeness and readability of the paper.

---

> > ### Comment · Reviewer_LGeA · 2024-08-12
> >
> > Thanks a lot for the rebuttal. I have raised the rating to 7.

---

> > > ### Author Response · Authors · 2024-08-12
> > > **Thanks for your efforts**
> > >
> > > We are delighted to know that our response has addressed your concerns and has led to an increase in your assessment score.
> > > Your recognition of the improvements made in the revised manuscript is highly appreciated, and it motivates us to continue refining our work to meet the highest standards of quality and clarity.

---

### Author Rebuttal · Authors · 2024-08-07

We sincerely appreciate all the reviewers for their valuable comments. Your feedback has helped us improve the quality of the paper and strengthen the arguments. We are pleased that most reviewers have a positive attitude towards our work :).

We are very grateful to the reviewers for acknowledging **our efforts in addressing important and challenging problems in long-tailed learning (Reviewers LGeA, qXeH, Vosp)**. We are also glad that they recognized **our innovations in theory and methods (Reviewers qXeH, Vosp), our comprehensive experiments (Reviewers LGeA, qXeH, VYLm, Vosp), and clear writing (reviewer LGeA)**. We have made our best efforts to respond to each suggestion, and we are very happy to further communicate if the reviewers have any questions. In addition, to deepen the reviewers' understanding of this paper, we will explain the interpretability and controllability of our method from the following perspectives:

**Perspective 1: Multi-expert structure enables different experts to focus on different classes**

This paper employs the same three experts with different loss functions as SADE: the forward expert tends to adapt to the long-tailed distribution similar to the training set. The uniform expert tends to adapt to the distribution with balanced classes. The reverse expert tends to adapt to the reverse long-tailed distribution with fewer head classes and more tail classes.

Through theoretical analysis, we can obtain the optimal solution for each expert, which corresponds to different class distributions:

- Forward expert:
$v_{1}^{*} (x) = \arg \max_{v_{1}} \mathbb{E}_ {(x,y) \sim p_{train}(x,y)} [ \log p(y\|x; v_{1}) ]$

- Uniform expert:
$v_2^*(x) = \arg\max_{v_2} \mathbb{E}_ {(x,y) \sim p_{uniform}(y)p_{train}(x|y)} [\log p(y|x; v_2)]$

- Reverse expert:
$v_3^*(x) = \arg\max_{v_3} \mathbb{E}_ {(x,y) \sim p_{inv}(y)p_{train}(x|y)} [\log p(y|x; v_3)]$

These results demonstrate that the multi-expert structure indeed enables different experts to focus on different classes, providing a foundation for the model to adapt to different distributions.

**Perspective 2: Dirichlet sampling and hypernetwork enable the model to adapt to different class distributions**

During the training process, we sample the weight vector $\boldsymbol{\alpha}=(\alpha_1, \alpha_2, \alpha_3)$ from the Dirichlet distribution $p(\boldsymbol{\alpha}; \boldsymbol{\beta})$, and weight the outputs of the three experts:

$$v(x) = \sum_{i=1}^3 \alpha_i v_i(x)$$

By adjusting the parameters of the Dirichlet distribution $\boldsymbol{\beta}$, we can control the distribution of the sampled weight vector to simulate different class distributions:
- When $\beta_1 > \beta_2 = \beta_3$, simulate the long-tailed distribution.
- When $\beta_1 = \beta_2 = \beta_3$, simulate the uniform distribution.
- When $\beta_1 < \beta_2 = \beta_3$, simulate the reverse long-tailed distribution.

Moreover, we introduce a hypernetwork $h_{\boldsymbol{\phi}}(\boldsymbol{\alpha})$ to map the sampled weight vector to the parameters of the experts:

$$\boldsymbol{\theta}_ {i} = h_{\boldsymbol{\phi}}(\boldsymbol{\alpha})_i, \quad i = 1, 2, 3$$

By optimizing the hypernetwork to minimize the expected loss:

$$\min_{\boldsymbol{\phi}} \mathbb{E}_ {\boldsymbol{\alpha} \sim p(\boldsymbol{\alpha}; \boldsymbol{\beta})} \left[ \frac{1}{n_s} \sum_{(x, y) \in D_s} \ell(y, v(x; h_{\boldsymbol{\phi}}(\boldsymbol{\alpha}))) \right]$$

We can obtain a set of expert parameters that perform well under different weight vectors. From an optimization perspective, this is equivalent to minimizing the expected loss of the ensemble model on the test distribution:

$$\min_{\boldsymbol{\phi}} \mathbb{E}_ {\boldsymbol{\alpha} \sim p(\boldsymbol{\alpha}; \boldsymbol{\beta})} \left[ \mathbb{E}_ {(x, y) \sim p_{test}(x, y)} \left[ \ell(y, v(x; h_{\boldsymbol{\phi}}(\boldsymbol{\alpha}))) \right] \right]$$

This shows that through Dirichlet sampling and hypernetwork optimization, we can obtain an ensemble model that adapts to various class distributions and performs well under different distributions. This adaptive ability enhances the robustness of the model, enabling it to cope with distribution changes in real-world scenarios.

**Perspective 3: Learning the entire frontier through Pareto optimization**

When optimizing the hypernetwork, we are actually learning the entire Pareto frontier. This is because we optimize the expected loss of the ensemble model under the weight vector sampled from the Dirichlet distribution. Different weight vectors correspond to different points on the frontier, representing different class distributions. By minimizing the expected loss, we are essentially balancing all these distributions and learning the entire Pareto frontier.

Theoretically, if a solution $\boldsymbol{\theta}(\boldsymbol{\alpha})$ is a local Pareto optimal solution to the expected loss optimization problem, then in a neighborhood of $\boldsymbol{\alpha}$, we can find a smooth mapping $\boldsymbol{\theta}(\boldsymbol{\alpha})$ that is Pareto optimal in the entire neighborhood. This result supports that our method can learn a set of continuous Pareto optimal solutions covering the entire Pareto frontier, providing flexibility to adapt to different practical needs.

**Perspective 4: Advantages over previous methods**

Compared with previous long-tailed learning methods, our method has several advantages in interpretability and controllability:
- Through the multi-expert structure and hypernetwork, our method can adapt to different class distributions.
- By learning the entire Pareto frontier, our method can control the behavior of the model by adjusting the weight vector to achieve dynamic adaptation to different class distributions. This controllability is not available in previous methods.
- We provide some relatively rigorous and coherent theoretical analyses to support the effectiveness of the method and enhance the interpretability of the model.

---

> ### Public Comment · ~Jinpeng_Zheng2 · 2025-02-23
>
> The $\mathbf{\alpha}\in\mathbb{R}_{+}^{k}$ in Eq 9 of the paper is inconsistent with the $\mathbf{\alpha}=(\alpha_1,\alpha_2,\alpha_3)$ here.

---

### Decision · Program_Chairs · 2024-09-25

**Decision:**

Accept (spotlight)

**Comment:**

This paper proposes a novel long-tailed learning method. It can generate a set of models to cover all distributions, and flexibly output a model that matches the user's preference. Empirical results validate the effectiveness of the proposed method.

The studied problem is an important and practical problem. The paper provides meaningful theoretical results and motivation for the new diversity expert learning method. The proposed method is novel enough. The paper provides an extensive set of experiments to support the claims.

In addition, the authors can further improve the paper according to the detailed reviews.